# Geophysical measurements of the Southernmost microglacier in Europe suggest permafrost occurrence in Pirin Mountains (Bulgaria)

Gergana Georgieva[1], Christian Tzankov[2], and Atanas Kisyov[2]

[1]Sofia University "Sv. Kliment Ohridski", Sofia, Bulgaria
[2]University of Mining and Geology "Sv. Ivan Rilski", Sofia, Bulgaria

**Correspondence:** Atanas Kisyov (at.kisyov@gmail.com)

**Abstract.** There are no large glaciers on the territory of Bulgaria but small patches of snow and firn have been observed in the high mountains at the end of summer. Perennial snow patches and microglaciers are considered as indicators of permafrost occurrence. The results from the first detailed geophysical investigations of Snezhnika glacieret, considered to be the southernmost microglacier in Europe, situated in the Golyam Kazan cirque, Pirin Mountains, Bulgaria, is presented in the paper.

Ground penetrating radar (GPR) and 2D Electrical Resistivity Tomography (ERT) were used to estimate the thickness of the microglacier as well as its subsurface structure. Measurements started in 2018 and continued over the next two years in order to assess changes in its size and thickness. The mean thickness of Snezhnika is about $4 - 6\ m$, reaching $8\ m$ or probably more in some areas. ERT measurements of the deeper parts of the microglacier beds show high electrical resistivities reaching over $60000\ \Omega m$ at a depth of $4 - 10\ m$. An anomaly at this depth is likewise distinguishable on the GPR profiles. These anomalies

are interpreted as permafrost areas and were consistently observed on the ERT and GPR profiles in the next two years of the study. These results imply for the first time the existence of permafrost in Pirin Mountains and in Bulgaria.

## 1   Introduction

82% of the glaciers in the world are smaller than $0.5\ km^2$ and cover 21% of the Earth's total glaciated area (Zemp, 2006).
Despite their small size, perennial snow patches and microglaciers are an important object of study for their vital role as water

reservoirs for many downstream ecosystems (Milner et al., 2009; Barry et al., 2011). They are sensitive to climate change although they are less influenced by global changes than glaciers (Glazirin et al., 2004; Williams et al., 2022). Perennial snow patches survive as a result of stabilization processes of accumulation as avalanching and wind-drift snow (Grunewald et al., 2010) and ablation (solar radiation, shading, debris) (Glazirin et al., 2004). Together with microglaciers they also are important for estimating local permafrost areas in high mountains (Hughes, 2014, 2018).

There have been no large glaciers on the territory of Bulgaria since the end of the Pleistocene (Gachev, 2020), but small patches of snow and firn have been observed in the high parts of the Rila and Pirin Mountains at the end of the summer. Two microglaciers are also located in Pirin Mountains. Snezhnika is the most studied among them (Gruenewald and Scheithauer, 2008; Gachev et al., 2016). It is also called "glacieret" or microglacier due to observed moraines, indicating movements of

the ice mass (Gachev, 2016). It is the modern remains of the Vihren glacier in Pirin Mountains and at present it varies in size between $0.02\ km^2$ and $0.07\ km^2$ (Gachev, 2017a). The first measurements of its size were made in the 1960's (Popov, 1964) and systematic measurements have been conducted every year at the end of summer since 1994 (Gruenewald et al., 2008; Gachev, 2016). The size of the microglacier is well monitored but information about its thickness is sparse. In October 1957, Popov (1962) bored into the middle part of Snezhnika and reached ground at $8\ m$. He also estimated the structure as follows: the upper $80 - 100\ cm$ represent an icy layer or icy crust which is under direct influence of the surface temperatures. Beneath this layer there is firn consisting of grain sizes between $1$ and $2\ cm$, increasing with depth. At the end of summer 2006, depth measurements were carried out again and three boreholes were made. The depth in two of them was estimated to be $11\ m$ (Gruenewald et al., 2008). Geophysical measurements for estimation of the thickness and structure of microglacier were not carried out until 2018 (Georgieva et al., 2019; Onaca et al., 2022).

Glaciers and permafrost are well studied separately but the interaction between them is a topic of less publications (Harris et al., 2005). Permafrost is a section of the subsurface in which the temperature is continually below $0°C$ for at least two years (Washburn, 1979; Harris et al., 1988). The definition is exclusively based on the temperature regime, and thus permafrost can exist in any type of sediment or rock (Ingeman-Nielsen, 2005). Permafrost can be absent in areas where glaciers are present but the ice rich permafrost and ground ice can be formed in front of and beneath the glaciers (Harris et al., 2005). Mountain permafrost is a good indicator for climate change (Guodong et al., 1992; Fort, 2015). Thawing of permafrost decreases the stability of slopes and can affect infrastructure in mountain regions. The main factor affecting mountain permafrost is topography (Etzelmüller et al., 2009) and especially the topographic conditions influencing the incoming solar radiation. According to Gruber et al. (2009), there are two types of surface phenomena indicating the presence of permafrost in mountains - rock glaciers and other creep phenomena and hanging glaciers and ice faces. Damm et al. (2006) give geomorphological indicators for mountain permafrost, among which are perennial snow patches. Perennial snow patches contribute to local permafrost occurrence and aggradation because they work as a shield with relatively high albedo, protecting the frozen underground from heat flux in summer. The existence of many perennial snow patches in an area in the mountains indicates a wider distribution of permafrost, especially in the shade of surrounding peaks (Haeberli, 1975; Rolshoven, 1982). Although there are high mountains reaching almost 3000 m in Bulgaria with suitable conditions for the presence of permafrost (Dobinski, 2005), there is a small number of studies on this topic (Onaca et al., 2020). No publications have been found to investigate the long-term state of the frozen subsurface in high mountains in Bulgaria.

With the development of modern technologies and in particular the equipment used in exploration geophysics, high-quality in-depth information can now be obtained. Geophysical techniques such as electrical resistivity tomography (ERT), and ground-penetrating radar (GPR) are widely used today for a multi-dimensional investigation of subsurface conditions in permafrost environments and corresponding landforms (Onaca et al., 2022; Navarro et al., 2009, etc). Until the late 1980's, they were mostly applied in polar regions (see the review by Scott et al. (1990)). The ERT technique is a commonly used geophysical method for permafrost evidence and studies (Mares et al., 1984, Scott et al., 1990). Geophysical methods are applied also in glacial studies with the GPR technique used for the imaging of glacial subsurface conditions (Navarro et al., 2009) and internal glacial structure (Arcone, 1996).

The application of geophysical methods in mountain permafrost regions is related to changes of the physical properties of earth material mainly associated with the freezing of incorporated water. The degree of change in the physical properties depends on water content, pore size, pore water chemistry, ground temperature and pressure on the material (Hoekstra et al., 1973; Hoekstra et al., 1974; King, 1984; King et al., 1988; Scott et al., 1990). When applied on permafrost, most geophysical methods detect parameters correlated to ice content (Hauck, 2001) like high electrical resistivities (Yakupov, 1973).

Using GPR and ERT methods, the structure, depth and extent of frozen areas beneath and near the snow field can be determined relatively quickly and easily, as well as the location of accompanying snow bodies and karst formations (Scott et al., 1990; Dimovski et al., 2015; Kisyov et al., 2018; Georgieva et al., 2019).

GPR is a high-resolution geophysical technique based on the propagation of electromagnetic waves. Dry snow and ice provide the optimal permittivity conditions of the radar signal from all possible geological environments for georadar pulses with a main frequency above $1\ MHz$. This is due to the extremely low degree of signal attenuation, which is a result of the low conductivity and the absence of any dielectric or magnetic relaxation processes above this frequency. The successful application of GPR in glaciology is related to the peculiar dielectric properties of frozen materials and to the big contrast with other geological materials (Evans, 1965; Fitzgerald et al., 1975). Employing high-frequency GPR antennas ($> 400\ MHz$) good results can be obtained in delineating the internal structure of glaciers or permafrost zones (Annan et al., 1976; Berthling et al., 2000; Hinkel et al., 2001; Jørgensen et al., 2007; Gadek et al., 2008; Onaca et al., 2015). The advantages of GPR are that data acquisition and processing are relatively fast and the interpretation result can be focused on different depths and scales, with a variety of antenna configurations and frequencies (Pipan et al., 1999; 2000; van der Kruk et al., 2003; Jol, 2009; Zhao et al., 2015; 2016).

On the other hand, snow and ice are an ideal environment for exploration, as stratigraphically they are made up of horizons with good endurance and characteristic shapes. The glaciers from temperate continental belts may contain layers rich in dust, sand, or rock debris from a few millimeters to tens of meters (Arcone et al., 1995; Lawson et al., 1998). The microglaciers and perennial snow patches in high mountains also contain layers of rock material between thin layers of snow, ice and firn (Kawashima et al., 1993).

ERT method can be successfully applied for studying permafrost areas (Kneisel et al., 2008). ERT is sensitive to the changes in the electrical properties of rocks, using different electrode circuits both vertically and horizontally (Dimovski et al., 2007). Changes in electrical resistivity depend not only on changes in lithology but also on the presence of water (Hoekstra et al., 1973; Olhoeft, 1978; Dimovski et al., 2015) in the cracks and pores of the rocks and its mineralization (Mares et al., 1984). The decrease in temperature leads to a decrease in electrolytic activity and hence a decrease in conductivity. This effect is significant below the freezing point and the resistivities increase exponentially (Hauck, 2001). A marked increase in resistivity at the freezing point was shown in several field studies (Ikeda, 2006; Kneisel et al., 2007; Mauer et al., 2007; Kneisel et al., 2008; Hilbich et al., 2009; Hausmann et al., 2012; Hauck, 2013; Supper et al., 2014; Onaca et al., 2015; Emmert et al., 2017). In most permafrost materials, electrolytic conduction takes place, where the current is carried by ions in the pore fluids of the material. In poor conductors with few carriers, such as ice, a slight displacement of electrons with respect to their nuclei produces a dielectric polarization of the material, leading to displacement currents (Telford et al., 1990).

ERT resulting subsurface plots suffer inherently from non-uniqueness as different subsurface features can have similar resistivities. GPR, as an independent method of subsurface investigation, can supply more accurate information about where the boundaries between layers are located. This in turn can be used to give some geological context to the ERT tomograms or even constrain the ERT inversion results. However, there are some limitations in using GPR and ERT methods in mountain regions and highly rugged terrain (Kneisel et al., 2008). A big limitation is the accessibility to the study site, the complicated logistics of transporting the measuring equipment, and the aspects regarding a safe system of work. Highly rugged terrain complicates the GPR data processing (Annan, 1999) and needs complex mathematical corrections for relief and proper estimation of the velocity of pulse propagation. The velocity is dependent on the dielectric properties of the media. Particularly in glaciers and permafrost it is a function of the temperature and water, ice, and air content. The above-mentioned inaccuracies can cause distortions in the spatial distribution of georadar data, registered boundaries, and, accordingly, the incorrect shape of the boundaries, the thickness of the individual layers and their actual depth. To minimize the inaccuracies the relief can be estimated using, for example, unmanned aerial vehicle (UAV) (Turner et al. , 2012; Sanz-Ablanedo et al., 2018). The velocities can be estimated using reference values from boreholes or by producing a velocity model based on the radargrams. ERT has limitations when applied in sites covered with coarse gravel or when a thick ice layer is present (Kneisel et al., 2008). Firstly, the effect from coarse gravel is bad contact between the electrodes and the ground, while in the second case the high electrical resistivity of ice can hinder the measurements. Distortions in estimating subsurface structure can be met in highly rugged terrain due to inaccurate estimations of the surface and measurement geometry. Interpretation of ERT and GPR data can also be ambiguous and consequently, it is advisable to use more geophysical methods on one site (Hauck, 2001; Kneisel et al., 2008).

In 2018, 2019, and 2020 geophysical measurements including GPR, 2D ERT, a very short meteorological record and surface capture with an UAV, were conducted by the authors and a team of students in Golyam Kazan cirque, Pirin Mountains. The aim was to investigate the thickness, internal structure of the Snezhnika microglacier, and the subsurface structure near it (within the glacial bed). In addition, we investigated where the meltwater from Snezhnika disappears beneath the microglaciers' bed. Even though this glacieret has been monitored since 1994 (Gruenewald et al., 2008) there is not much information about the ice thickness. It is necessary to know the thickness of the microglacier at many points to estimate the ice mass. Further measurements of its size and thickness will allow to monitor the mass balance of Snezhnika and then determine the relationship between the change in meteorological parameters and the change in ice mass.

## 2  Methods

### 2.1  Study site description

Pirin Mountains is a crystalline horst which is part of the Rila-Rhodope massif located in southwestern Bulgaria. The studied site is situated in the Northern Pirin, around the highest peak Vihren (2914 m). This part of Pirin Mountains consists mainly of marble that makes up the steep ridges and lends the relief its characteristic appearance (Boyadjiev, 1959). It has been subject to cryogenesis and karstification since the glaciers retreat (Gachev, 2017b; Gachev et al., 2019).

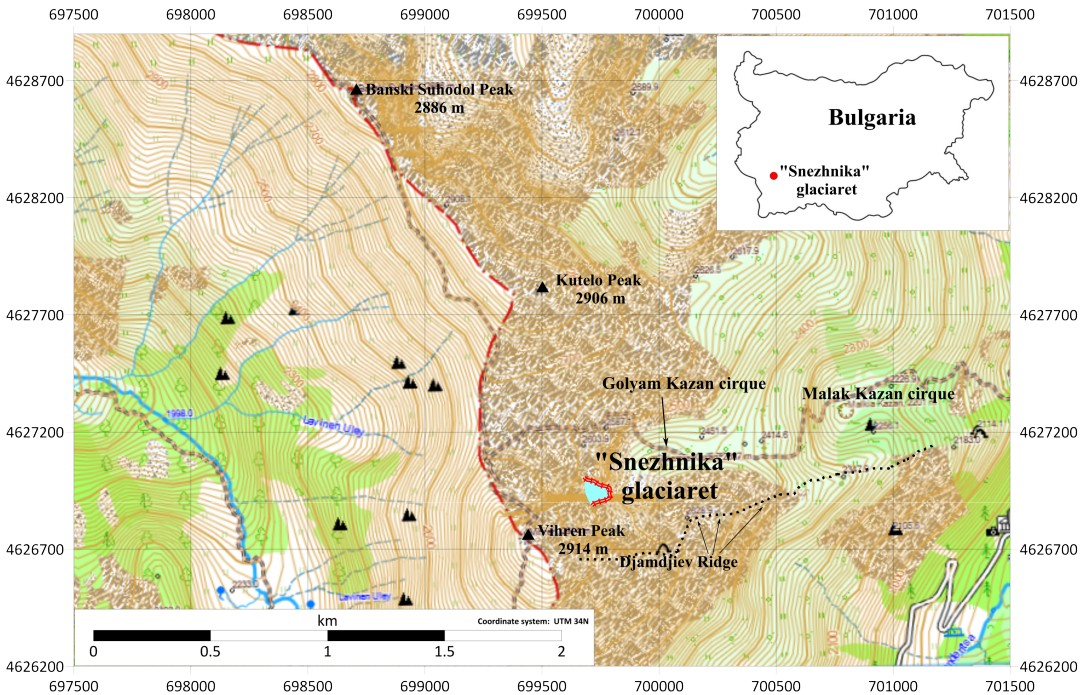

**Figure 1.** Snezhnika microglacier in Golyam Kazan cirque, surrounded by the highest peaks in Pirin Mountains - Vihren and Kutelo. On the northern side of the cirque is Dzhamdzhiev ridge, which starts from the Vihren peak and ends in the valley near the Banderitsa hut. Map base layer is provided by karta.bg.

The Snezhnika microglacier (Gruenewald et al., 2008), noted as the southernmost microglacier in Europe (Grunewald et al., 2010; Gachev, 2017a), is situated in the Golyam Kazan cirque. The location of Snezhnika is determined by the morphological features of the Golyam Kazan cirque, formed between the eastern slope of Vihren Peak and the southern slope of Kutelo Peak (2908 m) (Fig. 1), at around 2400 m asl. It is open to the east, with dimensions of $1200\ m$ by $1250\ m$ and a surface of about $1.2\ km^2$. The western and southern parts of the cirque are outlined by the steeper slope of Vihren Peak and Dzhamdzhiev ridge. The morphology of this slope, and in particular the morphology of the Vihren wall, at the base of which the glacieret is located, favors the accumulation of snow masses through avalanches and shading. The Vihren wall rises west of the microglacier and 420 m above its surface. The wall has mainly eastern exposure and partly northern exposure with slopes from $55°$ to $65°$ (almost half of its area). The largest slopes, reaching in places $85°$ to $90°$, are characteristic of the lower part of the wall. The steep slopes were formed during the final phase of the last (Wurm) glaciation (Popov, 1962, 1964).

Pirin Mountains is on the cross-road of Mediterranean and Continental climate. Snow cover in the mountains of southwestern Bulgaria is present for 180-200 days annually. The mean maximum thickness of snow is about 180 cm and the absolute maximum thickness measured until 2005 is about 350 cm (Brown et al., 2007). In the Golyam Kazan Cirque there is low coverage alpine and subalpine vegetation. It is present mainly in the central part of the cirque and is almost absent in the area of the microglacier. There is no evidence of surface water, lakes or rivers, in the cirque.

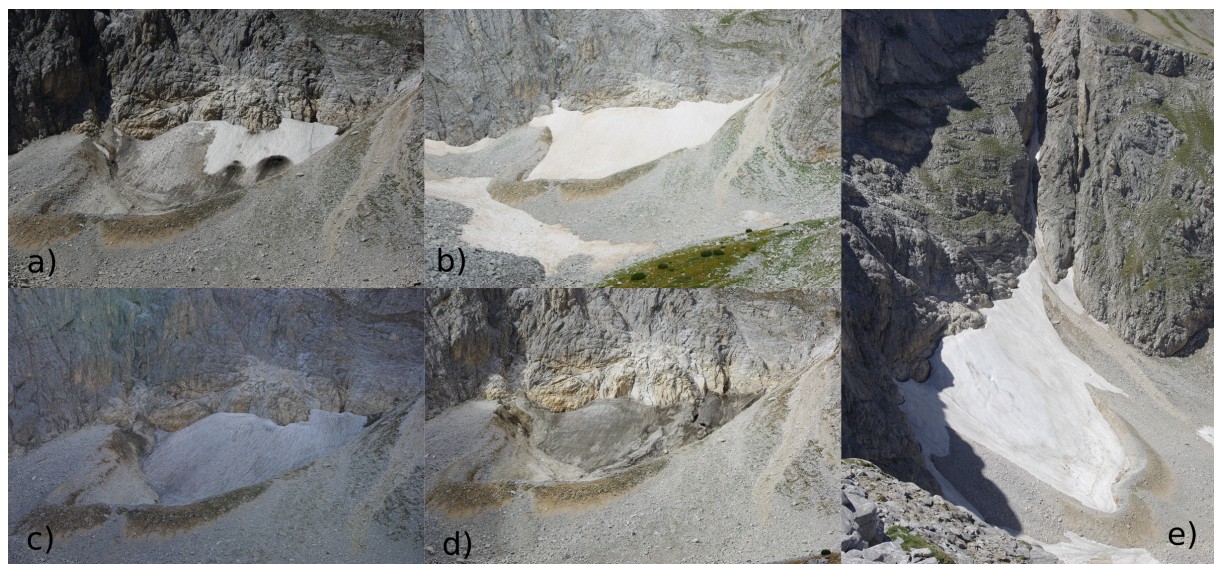

**Figure 2.** Snezhnika microglacier in Golyam Kazan over the years (a) August 25, 2017; (b) August 25, 2018; (c) October 5, 2019 and (d) October 9, 2020. Picture e) is taken from the Dzhamdzhiev ridge (near Vihren peak) on 01 July 2017. The shading of the ridge is already visible.

The size of Snezhnika microglacier varies from year to year as it can be observed in Fig. 2 but no trend can be determined (Gachev, 2016). Therefore, it cannot be said whether its size decreases or increases starting from the first measurements in 1994 (Gachev, 2016). During the period in which the present study was conducted (2018-2020) the size of Snezhnika decreased. This made it possible to make ERT profiles in the same place over the microglacier's bed where the GPR profiles has been made the previous year.

## 2.2 Ground-penetrating radar

The measurements in Pirin Mountains were carried out using a GPR system, including a SIR-3000 control unit and $270\ MHz$ antenna model 5104A by GSSI, Inc. USA. The settings used are listed in Table 1. By default, all radargrams are processed with the following processing levels:

- Pre-processing and geometrization of the radargrams (profiles lengths adjustments; orientation flip; declipping of extreme values and multiplying by scaling factor 1.44; dewow by subtracting mean value at 4 ns time window; resampling the data in x-direction by 2.5 cm trace increment; fixing the zero level by cutting the time section where waves pass through the air before the ground surface)

- Standard filtration and smoothing (cosine-tapered bandpass filter with low cut frequency 50 MHz, lower plateau 75 MHz, upper plateau 550 Mhz and high cut frequency 750 MHz; 2d median xy-filter on 5 traces by 5 samples window)

**Table 1.** GPR Settings of the measurements.

| SURVEY PARAMETERS | 2018 | 2020 |
|---|---|---|
| Scans per meter | 40 | 20 |
| Samples per scan | 1024 | 512 |
| Time window | 210 ns | 300 ns |
| Automatic gain | 4 pts | 4 pts |

IIR Filters Vertical: LP = 700 MHz HP = 75 MHz

– Signal amplification (profiles normalizing by energy equalization of the parallel profiles; profile trace normalize in order to produce mean amplitude equality distribution for all traces)

– Eliminate horizontal reverberations (applying a background removal filter)

The following generalized lithological media with their estimated mean velocities taken from standard properties tables were used for inverting and interpreting of GPR data in the radargram models: ice 0.15 $m/ns$, gravel with ice 0.13 $m/ns$, limestone 0.12 $m/ns$ (Baker et al., 2007).

GPR profiles within this study were made in 2018 and 2020. The locations are presented in Fig. 3. The profile coordinates and the outlines of the microglacier in the different years were recorded with Garmin GPSMAP 64st and Garmin GPSMAP 66s with error in horizontal coordinates $1 - 2$ $m$. The exact positions of the profiles and the outline of Snezhnika from 2018 were then evaluated from UAV images and the accuracy was improved. All profiles from 2018 are perpendicular to the slope and follow the relief's horizontals. The first one - GPR(2018)-01, is located in the lowest and comparatively flattened section of the microglacier, and the last, GPR(2018)-05, was in the highest elevation area accessible with the gear and tools available at the time. Two profiles from 2020 were situated along the slope and two were in the lower part of the microglacier's bed between head moraine and ice. The two profiles along the slope (GPR(2020)-1 and GPR(2020)-2) allowed determining the depth of the ice in the upper parts of the glacieret. There was no repetition of the GPR profile within the two years. Efforts were made to cover more microglacial area with GPR data and to estimate its thickness in more places. In addition, most of the GPR profiles from 2018 were not a part of the microglacier in 2020.

## 2.3 Electrical resistivity method

Based on other surveys conducted in different areas of Europe (Hauck, 2001; Ingeman-Nielsen, 2005; Mauer et al., 2007; Supper et al., 2014), the most suitable parameters of the measuring scheme were determined for optimal results when observing permafrost. Measurements in Pirin Mountains were carried out using 24 electrodes connected to a resistivity meter ABEM SAS 1000. The field measurements were performed utilizing a four-electrode Schlumberger array. The multi-electrode resistivity technique consists in using a multi-core cable with 24 conductors, as electrodes are plugged into the ground at a fixed spacing. The number of 24 electrodes allowed 121 measurements per profile. The relays which ensure the switching of those electrodes

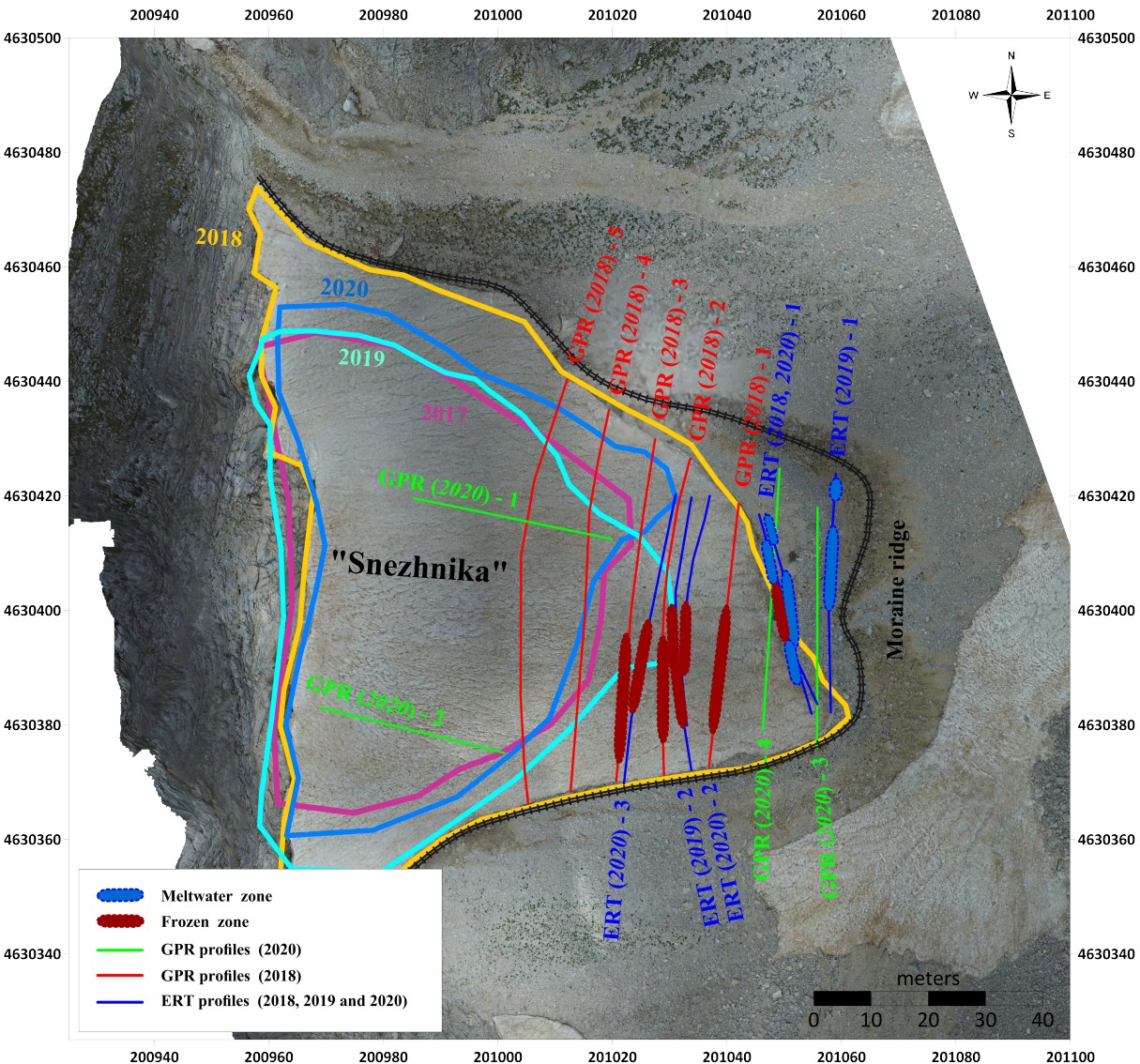

**Figure 3.** Location of the all GPR (red for 2018 and green for 2020) and ERT (blue) profiles in Snezhnika area. The size of the glacieret between 2017 and 2020 is given with lines in different colors. Bold red lines show the position of permafrost area, and bold blue lines show place of melt water (subsurface drainage system). The background picture, as well as moraine ridge outline (black line), is made with the UAV in 2018 in the same day of measurements.

according to a sequence of readings predefined and stored in the internal memory of the equipment are located in the resistivity meter itself. The various combinations of transmitting (A, B) and receiving (M, N) pairs of electrodes construct the mixed sounding/profiling section, with a maximum investigation depth which mainly depends on the total length of the cable. The lengths of the profiles were chosen to have the maximum length and depth of study, while not having to cross over the moraine

ridge of the microglacier (which was unstable and dangerous for climbing). Profiles are around $30 - 40\ m$ long with north-south direction and electrode spacing of $1.5\ m$. The depth of the study depends on the distance between the electrodes and the geometry of the circuit, the gradual increase of distance makes it possible to increase the depth of the subsurface study. In our study, a maximum depth of $9 - 10\ m$ is reached, and for most of the profiles the expected depth at this length of the measuring line with the multi-electrode system was $6 - 7\ m$. The selected length of the profiles and the type of measuring scheme were in accordance with the conditions in the field (namely, the width of the glacier bed and the safe access to the first and last electrodes).

The ERT data obtained was processed with RES2DINV (Loke, 2001, 2010). Data preprocessing includes extermination of bad data points and applying of a vertical/horizontal filter weight. An option for effect reduction of side blocks and obtaining smooth anomalies is also used. First, a trial inversion is made and afterwards, a RMS (Root Mean Square) error statistics is performed. Then, the data points with extreme values are removed and the inversion is made again.

ERT measurements near Snezhnika were conducted in 2018, 2019 and in 2020 along one, two and three profiles, respectively. They were situated in the lower part of the microglacier's bed on an area without any ice and snow cover (depending on the size of microglacier in the respective year Fig. 2). Thus, the first profile was measured over the three years, the second twice and the third - only in 2020. The last profile crosses a small part of the microglacier (Fig. 3). A pseudo-section of the apparent resistivity for each profile was then obtained. It consists of the measured values at certain points along horizontal lines at a certain depth. In order to obtain a model of electrical resistivity, an inversion algorithm proposed by Loke et al. (1996), was used in which a starting resistivity model is iteratively adjusted in order to achieve the best fit with the measured apparent resistivity values.

In addition to both main geophysical methods a digital terrain model (DTM) of Snezhnika with resolution 7.63 cm/pix was produced using UAV photogrammetry. The elevations in the study are taken from topographical maps in a scale 1:5000, from produced DTM and the GPS data.

Measurements were made on 25 August 2018, 4-5 October 2019, and 8-9 October 2020. Harsh weather conditions in the mountain prevented the taking of measurements on the same date every year. The time of measurements was chosen to be at the end of the summer and before the first snow in the mountain when the size of the glacieret is expected to be the smallest. This measurement window is relative, changing from one year to another, with 2017 and 2018 seeing the first snow fall in September, whereas in 2020, with a warm month of October, the first snow fall was recorded by the end of November.

## 3 Results and discussion

### 3.1 Thickness of Snezhnika microglacier

The thickness and internal structure of the Snezhnika microglacier were investigated using only GPR. Nine radargrams, were obtained, analyzed and interpreted and seven of them are presented in Fig. 4 and Fig. 5. Across these radargrams the discontinuity between the microglacier's ice and bed is well visible. On the other two profiles, situated in an area free from ice, this discontinuity is not presented, respectively.

In Fig. 4 are presented radargrams from 2018, which are horizontal relative to the slope. The uppermost layer represents the microglacier. Its depth varies between $1-2$ $m$ in the lowest part (Fig. 4a) and $5-6$ $m$ on the last profile (Fig. 4e). In 2018 the size of microglacier was bigger than in 2017 (Fig. 2a and Fig. 3) and accordingly, its lowest part consists only of the snow left from the last winter. This can be observed on profiles GPR(2018)-1, GPR(2018)-2 and partially GPR(2018)-3 (Fig. 4a,b and c) situated in the lowest part of the microglacier. Within the first layer, there are some less differentiated discontinuities. Probably, they are related to the periods of snow accumulation in winter, avalanches and periods of warm and cold weather during the winter of 2017/2018 when the melting and freezing occurred forming thin ice crusts.

The second layer lies under the snow and the ice, representing the glacial bed which consists of cobbles and pebbles. The voids between them are filled with water and this area has low resistivity values on ERT profiles. It is presumed that this layer drains the melted glacial water. The thickness of this layer varies from 1 to 4 $m$ along the particular lines and its depth is between $3.5-6$ $m$ on the first profile (Fig. 4a) and $5-8$ $m$ on the last one (Fig. 4e).

The third layer has a complex topography in the lower elevation survey lines, while in the higher elevation lines (GPR(2018)-03, GPR(2018)-04 and GPR(2018)-05, Fig. 4 c,d and e) it becomes almost parallel to the one above with a thickness of $4-6$ $m$. In the first and second profiles (GPR(2018)-01 and GPR(2018)-02), the layer is relatively thin in its central part (about 1 $m$), while two pocket-like recesses with thickness of up to about $6-7$ $m$ (and depth $10-12$ $m$ from the ground level) are formed along the left part. The presence of ice causes a decrease of reflections of electromagnetic waves and based on this it is assumed that the pocket-like structures are ice lenses or ice-rich area.

Pale rectangular areas are visible on the radargrams. They are most visible on the higher elevation profile (GPR(2018)-05, Fig. 4e). They have no scientific meaning and are the result of technical difficulties due to the steep slope in the upper part of the microglacier. The measurements were stopped and resumen several times and this caused gain level changes in some places.

In 2020 two more GPR profiles were made on Snezhnika in order to add more information about the thickness of the ice. The first profile (GPR(2020)-1, Fig. 5a) clearly outlines the lower surface of the glacieret with a depth of between 4 and 7 $m$. At the beginning of the profile (left part of radargram), fading of the phases is observed at a greater depth ($> 7$ $m$), which may be due to a frozen zone beneath the upper part of the microglacier. In the first half of the radargram a hill-like structure is visible, which is interpreted as a plucking zone of the microglacier. On the right part of the profile the discontinuity between the microglacier's ice and the bed has the greatest depth of 7 $m$, which becomes shallower at the end of the profile, in the last $4-5$ $m$ of it. Beneath the lowest part of the microglacier another hill-like structure is visible. It can be interpreted as the new head moraine.

In the second profile (GPR(2020)-2, Fig. 5b), the relief of the lower surface is clearly traced at a depth of about 8 $m$. Here, a layer with a depth of 4 to 8 $m$ is distinguished, which is composed of either frozen and well-joined rock blocks or older ice. At the lower end of the profile (right part of the radargram), single reflections can be observed in this layer, caused by boulders covered by ice.

The obtained depths of the Snezhnika in 2020 correlate very well with the results obtained in 2018. At that time, the depth in the uppermost profile is about $6-7$ $m$ (Fig. 4e). Profile GPR(2020)-1 intersects profiles GPR(2018)-4 and GPR(2018)-5 and as

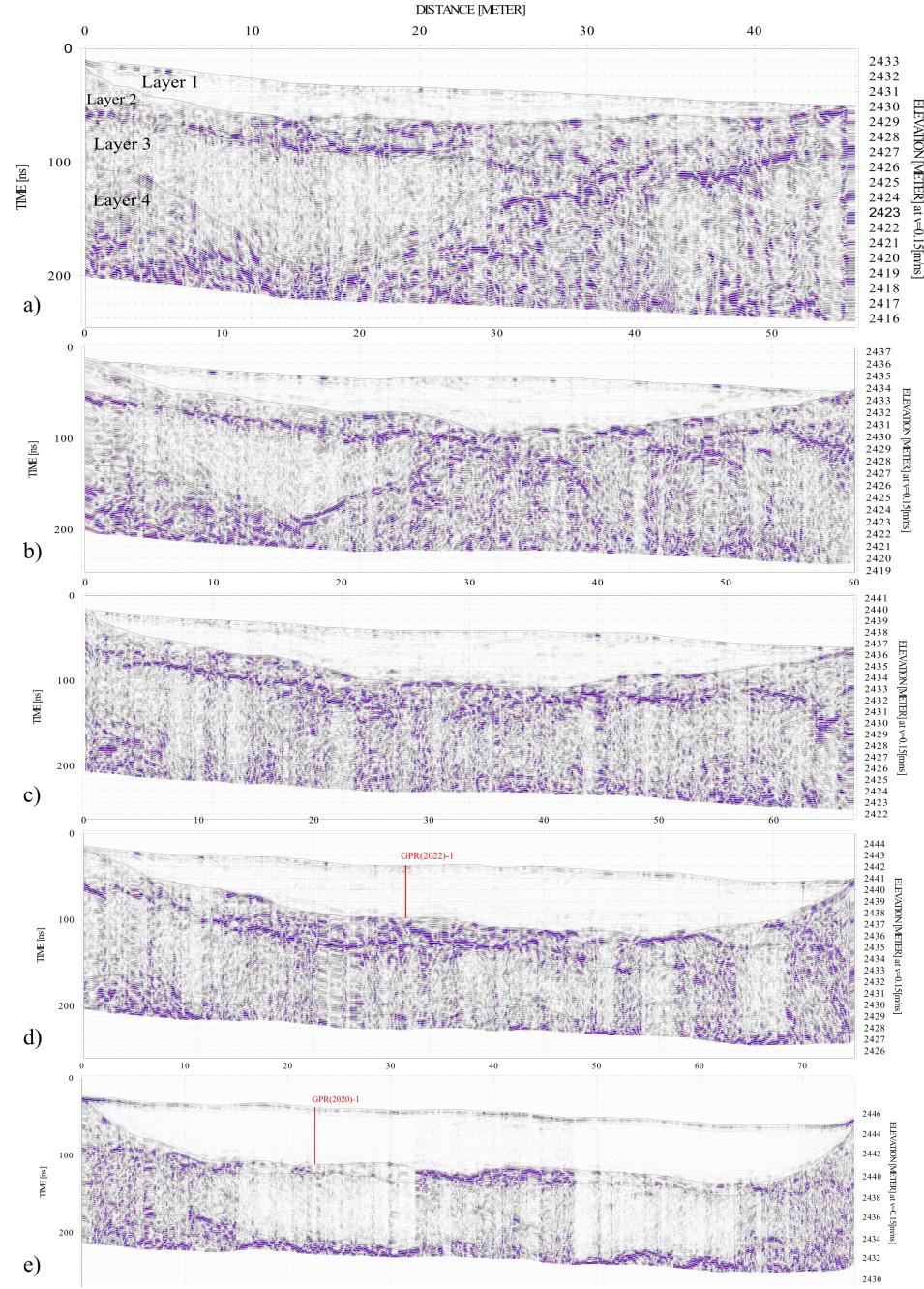

**Figure 4.** GPR profiles from 2018: a) GPR(2018)-1 situated at the lowest altitude; b) GPR(2018)-2; c) GPR(2018)-3; d) GPR(2018)-4; e) GPR(2018)-5 situated at the highest altitude. All profiles are with horizontal elevation. Red lines indicate the cross point with GPR(2020)-1.

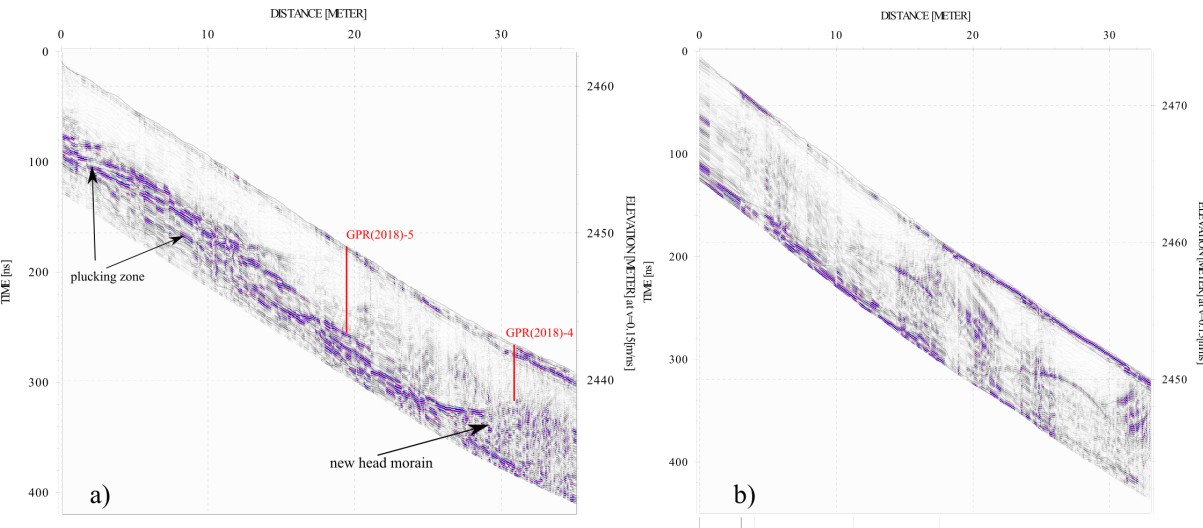

**Figure 5.** GPR profiles from 2020 along the slope of Snezhnika: a) GPR(2020)-1 north; b) GPR(2020)-2 south. Red lines indicate the cross point of GPR(2020)-1 with GPR(2018)-4 and GPR(2018)-5.

can be seen from Fig. 3 and Fig. 5a the place of intersection is also the place of the greatest thickness, with a depth of $7\ m$. The deepest part of the microglacier is detected on GPR(2020)-2, where the thickness of the ice is $8\ m$. This profile is located in the
southern part of microglacier, which is most shaded by Vihren wall and Dzhamdzhiev ridge (Fig. 2e). The effect of shading is also well visible on Fig. 4a where the snow layer is thicker in the southern part of the profile. The obtained maximum thickness of the microglacier shows agreement with the results from early borehole measurements conducted by Popov (1962). The depth of 11 m obtained by Gruenewald et al. (2008) is not detected. It should be noted that borehole measurements of depth give only point information, compared to the profile measurements conducted by the authors. Not the whole area of the microglacier was
covered by GPR profiles in years 2018 and 2020 and it can be a reason for not observing a depth greater than $8\ m$. Onaca et al. 2022 measured a depth of the border between ice and gravel of 12 m in a small area in the upper part of the microglacier, which is probably the maximum estimated thickness of Snezhnika. GPR profiles made within the present study do not cross this area. This indicates a necessity for using a thicker net of GPR profiles in the future to better map the lower border of the microglacier. The main layers outlined in the study area are also presented in Table 2.

**3.2 Subsurface structure of microglacier's bed**

The two GPR profiles from 2020, situated in the glacial bed and in the lower in elevation part of the investigated area (Fig. 3) were also covered by ERT measurements. Fig. 6 shows the electrical resistivity values for profile ERT-1, measured in 2018, 2019 and 2020. On the resulting plot, three zones can be clearly distinguished. Zone-1 is situated near the surface and represents a mix mainly of pebbles and cobbles. It is characterized by a relatively high electrical resistivity of $8000\ \Omega m$ to $40000\ \Omega m$.
Based on these values (Dortman, 1984) and the lithology of the area, it can be assumed that this layer consists of broken marble

**Table 2.** Description of the stratigraphy of the studied area.

| Layer | Depth | Thickness | Description | GPR velocity |
|---|---|---|---|---|
| 1. Microglacier | From $1-2\ m$ in the lowest part and up to $7-8\ m$ in the higher part | 2 to 8 $m$ | The top layer combines the inner sublayers and discontinuities of the microglacier | $0.15m/ns$ |
| 2. Glacial bed | Between $3.5-6\ m$ in the lowest part and $5-8\ m$ in the upper parts of the slope | 1 to 4 $m$ | Rock blocks of different size with voids and channels between them filled with ice and water | $0.13m/ns$ |
| 3. Permafrost | 7 to 9 $m$ in the lower parts of the relief with two pocket-like recesses up to $7-9\ m$ and between $11-13\ m$ in the higher parts | 1 to 8 $m$ | Permafrost zone with two ice lenses along the left and the right side under the glacial bed of the microglacier | $0.15m/ns$ |
| 4. Bedrock | $>14\ m$ | $>10\ m$ | Fractured marble rocks massif | $0.12m/ns$ |

rock. In 2018 (Fig. 6c) the thickness of this zone is $1.5-2\ m$, and in the next two years it reaches up to 4 m (Fig. 6a and b). Below the first zone, at a depth of 1 to 5 $m$, the second zone is located (Zone-2). It is characterized by relatively low values of the specific electrical resistivity within the range of $1000\ \Omega m$ to $8000\ \Omega m$. This zone represents a highly watered zone. Its size is smaller in 2018 and it is located mainly at the edges of the glacial bed at a depth of 3 $m$. In the next two years its size

increases and its thickness decreases to 2 $m$. The deepest zone (Zone-3) is of greatest interest and is characterized by resistivity over $60000\ \Omega m$. High resistivities (Hauck, 2001; Kneisel et al., 2008) are typical for ice and permafrost and respectively, Zone-3 represents an ice-rich permafrost area.

In 2018 the melt water was drained around the frozen subsurface areas in the lower part of Snezhnika in Zone-2. In 2019 and 2020 the size of the glacieret was smaller and this obstacle no longer existed or it was deeper than the depth reached by the ERT

method. Then the main flow of meltwater was directed below the central part of the microglacier's bed. Profile ERT(2019)-1 is situated 5-6 m lower at elevation from profiles ERT(2018)-1 and ERT(2020)-1 (Fig. 3). On this profile, the lowest resistivities in Zone-2 are observed. Probably the melt water is collected in this area in the lowest part of the microglacier's bed and then it flows deeper through a karst structure. This could be assumed because the area is very close to the head moraine, but no surface water is observed on the opposite side of the moraine or down along the slope.

In the second electrical profile (ERT(2019,2020)-2), presented on Fig. 7, two zones are distinguished. Zone-2 is located in nearsurface area of the profile, with a thickness of up to 2 $m$. This area represents the accommodating medium composed of crushed marble pieces of different sizes(as the marble is the main rock type in the area), having a resistivity between $10000\ \Omega m$ and $40000\ \Omega m$. The zone is highly watered and is a result from the melting of the microglacier. In the southern part of the profile at a depth beyond the third meter in the section Zone-3 is located. This zone appears with values of electrical resistivity

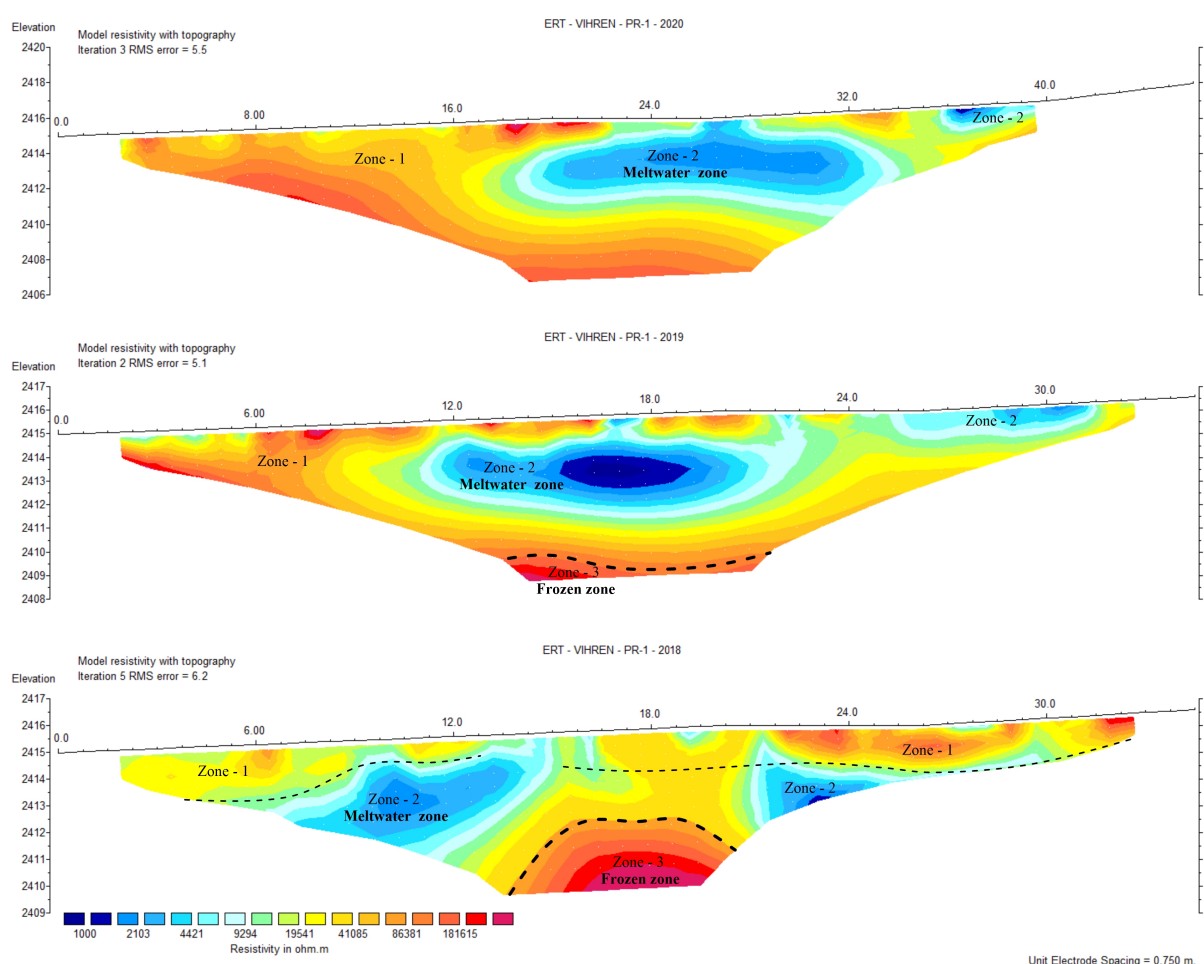

**Figure 6.** Electrical resistivity sections along ERT-1 obtained near Snezhnika microglacier in 2020 (a), 2019 (b) and 2018 (c). Blue color represents low resistivities and red color indicates areas with very high resistivities.

over 60000 $\Omega m$ and represents an ice-rich permafrost area in the base of the microglacier. In 2020 (Fig. 7 down) the frozen area is located about 1 $m$ deeper than in 2019 (Fig. 7 up). Probably the sinking of the permafrost area in 2020 compared to 2019 is a result of two factors. The first one is the interannual change of the meteorological parameters. In October 2019, there was much less precipitation in the area (below average) and slightly lower air temperatures than in October 2020, when air temperatures were higher and the precipitation was above average (based on Copernicus Climate Change Service reports). This might have led to an increase in the active layer thickness in 2020. The second reason for this change is that the two profiles do not exactly match in location. Although they are quite close to each other, the small displacement may be the reason for the greater change in depth.

Fig. 8 shows the third ERT profile (ERT(2020)-3), located just below the glacieret. This profile was measured only in 2020 when the size of Snezhnika was the smallest. The measuring line passed through a small piece of the microglacier, which is

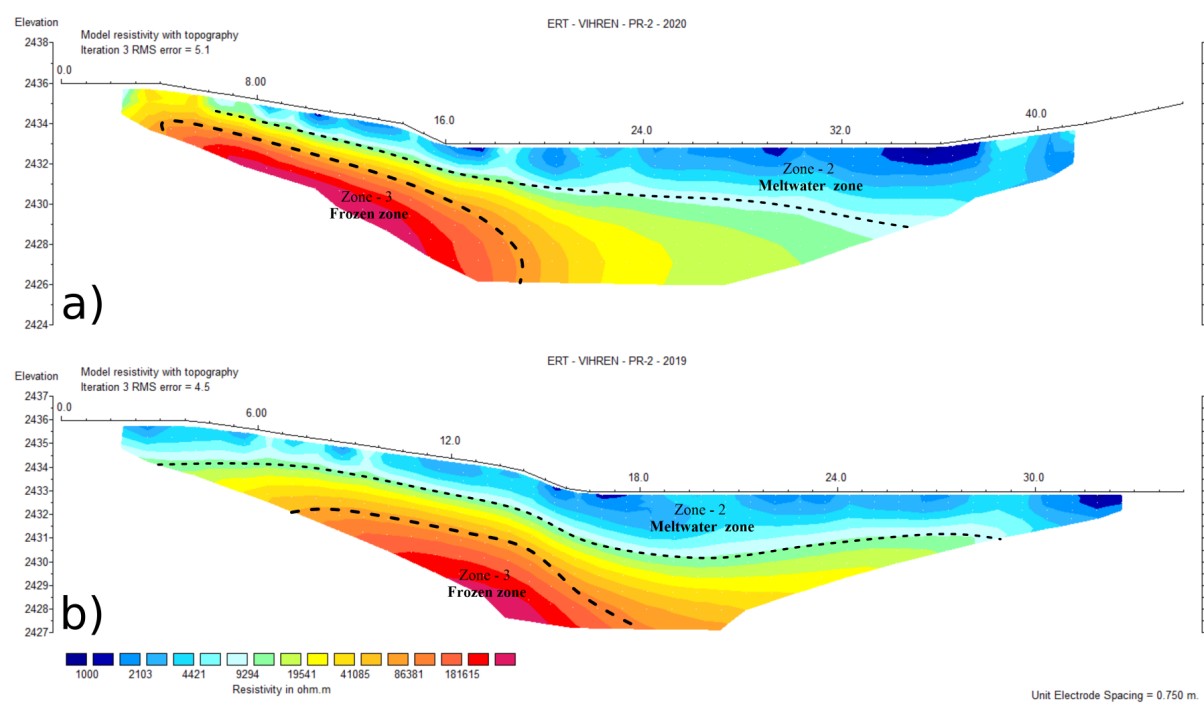

**Figure 7.** Electrical resistivity sections along ERT-2 obtained in Golyam Kazan area in 2020 (a) and 2019 (b).

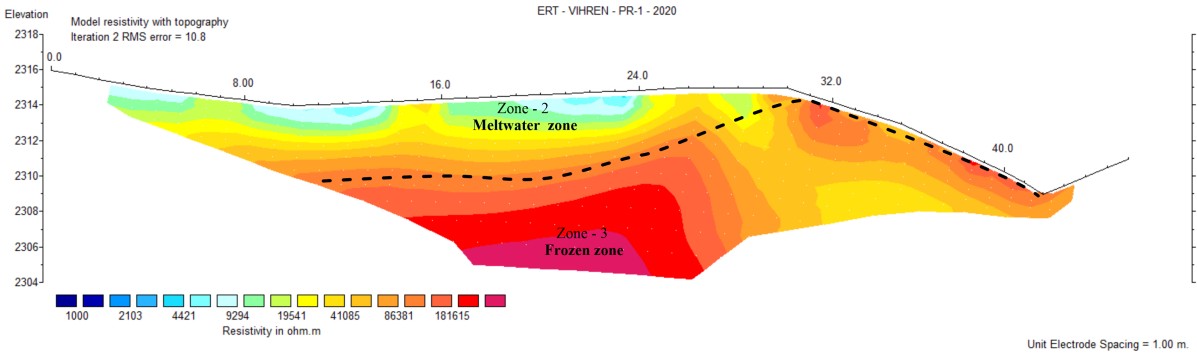

**Figure 8.** Electrical resistivity sections along ERT-3 obtained in Golyam Kazan area in 2020.

well seen on Figure 8. Only one electrode was in the ice, which was covered by a thin debris layer, and this probably made the measurement possible. Zone-3, representing again the ice-rich area in the base of the microglacier, occupies a large part of the section. It is located at a depth of $4\ m$ in the southern part of the profile and in the northern it reaches the surface. Namely, the northern part of the profile crosses part of the glacieret (Fig. 3). Zone-2 on ERT-3 is distinguished only in the very shallow parts of the profile.

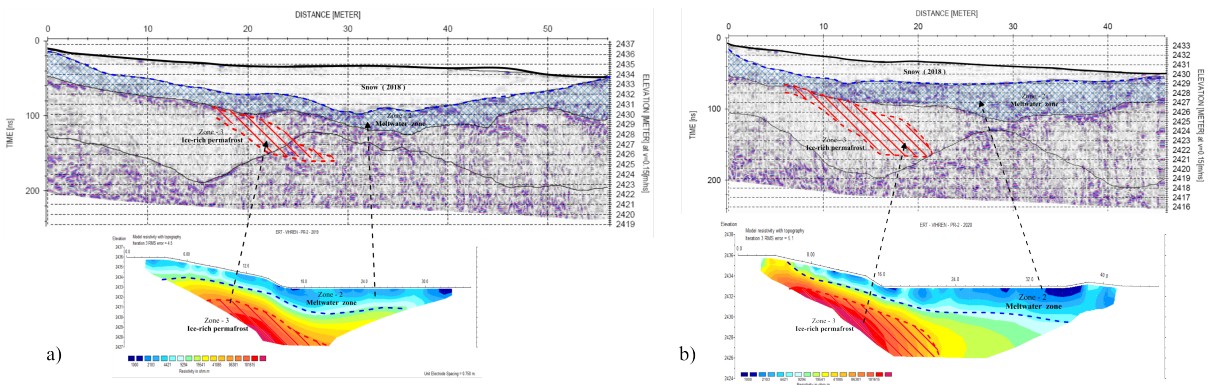

**Figure 9.** Aligned profiles ERT(2019)-2, ERT(2020)-2, GPR(2018)-1 and GPR(2018)-2. The red-shaded zones have the same size on ERT and GPR profiles to show areas of overlapping for permafrost zone. Blue-shaded areas present the layer draining the meltwater.

On the aligned GPR and ERT profiles shown in Figure 9 the permafrost area and drainage layer of melt water are better visible. The figure presents the alignment of profiles ERT(2020)-2 and GPR(2018)-1 and profiles ERT(2019)-2 and GPR(2018)-2. The two ERT profiles are located between GPR profiles and are shifted by several meters. The surface beneath the snow on the GPR profiles has a similar shape as the surface on the ERT profiles without snow. On the aligned plot the high resistivity zone fits well with the area identified as ice rich permafrost on the GPR profiles. This zone is observed on both GPR(2018)-1

and GPR(2018)-2. Over the next two years, ERT(2019)-2 and ERT(2020)-2 profiles show very high resistivities at the site where the permafrost zone was observed. In Figure 9, these zones of overlap are shown with a red-shaded area on radargrams. ERT profiles have shorter lengths and smaller depths and makes it possible to explore only part of the permafrost area. Some changes in the depth and expansion of the zone during the period (2018-2020) can also be observed. But as the profiles do not fully overlap, it is difficult to analyze the reasons for these changes and it is advisable this finding to be investigated in-depth

during future measurements in the area.

The study presented includes measurements made over three consecutive years and a frozen zone is observed every year. As permafrost is defined as subsurface area the temperature of which remains below $0°$ for at least two consecutive years (Harris et al., 1988) or in our case, with the presence of ice, this can be treated as evidence of permafrost existence in high mountains (particularly Pirin Mountains) in Bulgaria. The permafrost area near Snezhnika microglacier is presented by a zone with a lack

of reflections on GPR. This allows us to assume that this is an ice-rich permafrost. It might be buried ice, a remnant from Snezhnika when it had a bigger size for several consecutive years, followed by a long warm period of debris accumulation over the ice.

Our results show that the permafrost zone is situated mainly in the southern part of the microglacier's bed (Fig. 3). Also, the southern part of the microglacier is at least 1 m thicker than the northern one (Fig. 5). The snow layer in 2018 is also thicker

in the southern part (Fig. 4a). The reason is that this part is closer to Dzhamdzhiev ridge and the wall of the Vihren peak, both with very steep slopes and rising 300 - 400 m above the surface. They shade the Snezhnika for most of the day, as it is visible in

Fig 2e. The picture is taken from Dzhamdzhiev ridge on 1. July, in the middle of the day and the shadow over the southern part of the microglacier can already be observed. The shading effect causes the northern part of the area to be exposed to the sun longer than the southern part. During the summer this protects the microglacier and the permafrost area from increasing solar radiation. Proof of this assumption is the presence of snow patches, located at the bottom of the Dzhamdzhiev ridge, close to the Snezhnika. Snow patches are not observed in the northern part of the cirque. The role of shading from surrounding mountain ridge and the glacieret is very important for preservation of the ice rich area. The other factors for permafrost formations and preservation should also be investigated in future studies in order to have a more complex view on processes that keep the buried ice near the Snezhnika but also in similar locations worldwide.

The permafrost zone in Golyam Kazan cirque was obtained in 2018 on the ERT profile and in the GPR profiles PR(2018)-1 and GPR(2018)-2. The 2020 results show again indications of a frozen area below the surface, although its upper part was not found at the same depth as in 2018. One reason for this may be the smaller size of Snezhnika in 2020 compared to 2018. The glacieret preserves the frozen subsurface area in summer. Even when the size of the microglacier is smaller, a permafrost zone exists but is observed at a greater depth. This change in depth is probably due to several factors such as the lack of shading from glacieret, the interannual changes in meteorological parameters between 2018 and 2020, and the shift of the GPR and ERT profiles between the campaigns.

Bulgaria and the Balkan Peninsula are situated in lower latitudes where no continuous permafrost exists. Only isolated patches of permafrost are present in high mountains (Brown et al., 2001). However, its distribution is much less investigated (Oliva et al., 2018). Permafrost probably exists above $\sim 2350\ m$ in Rila Mountains (Oliva et al., 2018) and above $\sim 2400\ m$ in Pirin Mountains. In the Golyam Kazan cirque it can be sporadic and present in the area south of the glacieret, where snow patches are observed in some summers. To prove this more studies including geophysical measurements are needed. The area was described above as well shaded by the Dzhamdzhiev ridge. Most of the places in the high mountains of Bulgaria, where probably permafrost is present, are steep and this makes it difficult for geophysical measurements to be carried out. Nevertheless, more investigation into permafrost distribution and its monitoring in time should be made. It is important because changing climatic conditions can lead to the disappearing of isolated permafrost patches especially in areas with relatively warm climate like the Balkan Peninsula. Disappearing of permafrost patches can cause rapid degradation processes and rock avalanches. In Bulgaria there is not a lot of infrastructures in the high mountains that can be damaged, unlike in the Alps or other inhabited mountains. However, many hike trails cross slopes that can be unstable and dangerous when permafrost melts.

## 4 Conclusions

Detailed geophysical measurements of Snezhnika microglacier in Golyam Kazan, Pirin were conducted in 2018, 2019 and 2020 in order to estimate the thickness and internal structure of the glacieret and the subsurface structure beneath it. One of the main results from the study is a more comprehensive, large-scale assessment of the ice-thickness than it was done before. The mean thickness estimated from GPR profiles is about $4-6\ m$. In some places in the southern part of the ice body it reaches $8\ m$. This results show partial agreement with the results from early borehole measurements and results obtained by Popov

(1962) but the depths of 11 m (Gruenewald et al., 2008) and 12 m (Onaca et al., 2022) are not detected. The reason for this can be that our GPR profiles cover large but not the whole area of the microglacier. It is still not possible to estimate the changes in thickness of the microglacier through the years starting from 1962 due to insufficient data. However, the thickness values estimated are a good base for monitoring the microglacier Snezhnika not only by its surface area.

The second significant finding is the indication of permafrost in Pirin Mountains. The data from GPR and ERT measurements
which complement each other allow us to distinguish a zone of ice-rich permafrost in the southern part of the microglaciers bed, between the head moraine and the ice. The zone has a complex shape, lack of reflections in GPR profiles and very high resistivity, $\geqslant 60000\ \Omega m$, values typical for ice. ERT measurements were repeated over three consecutive years, detecting the anomaly during every measurement campaign. This can be taken as an evidence of permafrost in Pirin Mountains.

An important finding is also the obtained information on the hydrology of the microglacier's area. This is also important for
the preservation of snow and ice. Based on our observations, the hypothesis where does the meltwater disappear can be stated. It was identified that the underlying layer is most likely draining the melted glacial water. Microglacier drainage system is fully situated beneath the surface as surface water is rarely observed and in most cases it is only close to the glacieret.

The frozen zone is situated in the southern part of the microglacier and exists beneath the snow (in 2018) and without snow cover as in late autumn of 2020, when the microglacier's size was the smallest and no shading of the ice layer was possible. The
area is closer to the Dzhamdzhiev ridge and is shaded by it for the most of the day in summer. This result suggest the importance of mountain ridge shading for the preservation of frozen subsurface areas in the Golyam Kazan cirque, Pirin Mountains but the role of other factors should also be considered.

The knowledge of permafrost on the Balkan Peninsula and, particularly in Bulgaria is very general and studies of permafrost, including geophysical measurements, are still rare. Even though the present study is focused on a small area of Pirin Mountains
(Bulgaria), it gives important information on the presence, extent, and the state of a permafrost area which has survived due to the local conditions despite warming climate.

*Data availability.* Authors are happy to share data requested by email. Available data is: GPR data from Golyam Kazan (2018 and 2020); ERT measurements in Golyam Kazan (2018, 2019 and 2020); GPS tracks of snowfields, photos and short meteorological records of the temperature, atmospheric pressure and humidity made in two points in Golyam Kazan. DTM with resolution 7.63 cm/pix was also constructed
using UAV photogrammetry.

*Author contributions.* AK led ERT measurements and processed the data. CT led the GPR measurements and processed the data. GG led the projects and organized the field work. GG prepared the manuscript with contribution of all co-authors.

*Competing interests.* The authors declare that they have no conflict of interest.

*Acknowledgements.* The field work was supported by the Science Fund of Sofia University within the projects 80-10-126/21.04.2017, 80-10-217/26.04.2018 and 80-10-24/18.03.2020 and Science Fund of University of Mining and Geology within the project GPF-222/11.03.2019. The publication is funded within the framework of the National Science Program "Environmental Protection and Reduction of Risks of Adverse Events and Natural Disasters", approved by the Resolution of the Council of Ministers N 577/17.08.2018 and supported by the Ministry of Education and Science (MES) of Bulgaria (Agreement N D01-279/03.12.2021). All measurements were carried out with the participation of students from both universities. The authors thank the following students: Boriyana Chtirkova, Bojourka Georgieva, Daniel Ishlyamski, Dragomir Dragomirov, Yanko Ivanov, Spas Nikolov, Valentin Buchakchiev, Kalina Stoimenova and Angel Dimitrov. We thank also Vassil Gourev for sharing idea and knowledge of studying perennial snow patches in Bulgarian mountains. Measurements were performed with the permission of Pirin National Park administration.

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
