# Peer review of "Geophysical measurements of the Southernmost microglacier in Europe suggest permafrost occurrence in Pirin Mountains (Bulgaria)"

_The Cryosphere, 2021_

## Referee Comment (RC1)

Review of the manuscript entitled:

GEOPHYSICAL MEASUREMENTS OF PERENNIAL SNOW PATCHES IN PIRIN MOUNTAIN, BULGARIA

**General comments**

The research article enclosed aims to characterize perennial snow patches and the structure of their underlying formations in Pirin Mountains, Bulgaria, using geophysical methods, such as: Ground Penetrating Radar and Electrical Resistivity Tomography.

First of all, even though this type of characterization has been done before in other parts of the world, I believe it is novel and intriguing in this context, due to the rarity of microglaciers and perennial snows on the territory of Bulgaria, despite the high altitudes. I appreciate the hard work that has been put into this study. From personal experience, it is not easy to conduct geophysical surveys on mountainous terrain, especially if it is not easily accessible as this site appears to be. Normally, I would expect more repeatability of the surveys done throughout the years in order to capture the morphological/hydrological changes that might have occurred, but given the location, this may not have been an option.

However, I do have several problems with this manuscript and in this current state I would only recommend a reconsideration with major revisions. I do believe this work is publishable, but needs a bit more work on the following aspects:

1. The use of English language. Even though I often understand the message behind a sentence, it is very hard to follow sometimes. There are a lot of grammatical errors and quite often I come across sentences that could be reworded or incorporated in a longer phrase to improve the readability of the paragraph. (I will give specific examples below and in some places I attempted to rephrase some parts in order for them to read better. I should say that I only picked up some mistakes, there are definitely others that would need attention).

2. I feel like there needs to be a bit more discussion about the changes you might have seen across the 3 years of surveys. All around us, nature is changing rapidly and cold regions are often the ones that show the greatest change. You touch a bit upon some structural changes you noticed on the tomograms imaging the microglacier, but I personally feel this should be expanded enriching the content of the manuscript.

3. There are a few technical issues that I came across when reading your methods and looking at the figures, but I think these can be overlooked if only they are properly explained (I will give details below).

I will give a list of specific comments below with line numbers associated. The ones with red headings I considered to be more pressing issues that need resolving. When I use quotation marks I am rephrasing that bit of the text or correcting a mistake.

**Specific comments**

**Abstract**

1. Line 4: rephrase as "in Europe, situated in the...". This is an example of a sentence that could have been merged into a bigger phrase to improve readability.

2. Line 5: ERT stands for Electrical Resistivity Tomography

3. Line 6: "next three years" implies 2021 as well. Please rephrase.

4. Line 11: No issue here, just wanted to underline this is an interesting find. Like I said above, the permafrost's changes in time would be interesting to look at. This could be a continuation of this work, a basis for an ampler future project.

**1. Introduction**

Line 22: "The most studied perennial snow"

Line 25: "size were made"

Line 25: What systematic measurements ? Be specific please. Also, "were conducted".

Line 25: "at the end of summer"

Line 31: There are many types of geophysical measurements, not just the ones listed here. A probe reading a soil's temperature can be thought as a geophysical measurement. Are you sure none have been carried out before? You could say: "Georgievna et al., (2019) finds that geophysical measurements, such as... could be useful for a better site characterization because of their...".

Line 35: "Permafrost is also a good..."

Line 45: Double parenthesis after Onaca et al.

Line 52: " with the GPR technique used for imaging of..."

Line 58: ERT is not an acronym for electrical resistivity. ER would be, if you want to use one.

Line 63: I am not sure if you need the EM abbreviation if you don't use it afterwards.

Line 79: ERT is not just in 2D, can be in 1,2,3 D and even timelapse 3D classed as 4D in some cases. Please rephrase.

Line 81: Do not forget salinity!

Line 93: The last sentence is a repetition, please rephrase or remove. In addition, the last paragraph is a bit too short, normally is meant to be setting the scene to what you are about to read in the paper.

Please expand. For example you have not even mentioned which geophysical methods you have used, even though you are hinting that GPR and ERT are your methods of choice.

## 2. Methods

Line 97: You can just say ERT from now on. It was previously defined in the abstract and introduction and I suspect you will be asked for a list of abbreviations too.

Line 98: I do not thing this intro paragraph is needed, at least not in this form. Everything you say here you repeat later on.

### 2.1

Line 113: "as can be observed in Fig.2"

Line 116: "in the same place over the microglacier's bed where the GPR profiles were made the previous year"

Figure 2 caption: "over the years".

Figure 2 caption: 2nd sentence should not be in a caption.

Figure 2: By looking at the dates when the photos were taken, can you discuss a bit if the weather at the specific time of year had an impact on your measurements? In 2017 and 2018 you went there in August, whereas in 2019 and 2020 you went in October.

### 2.2

Line 133: "Locations are presented in..."

Line 133: "Profile coordinates..."

Line 134: 1-2 m seems like a rather big error to me. Can you justify that this is acceptable?

Line 135: The "relief's horizontals" ?

Line 139: I really don't get why they have no topography, doesn't it skew the results? surely the topography is not that similar across all radargrams.

Line 141: Even though I sympathise with the effort needed to perform this kind of survey, unless you describe how this has impacted your results, this last sentence is unnecessary.

**2.3**

Line 144: "Based on other surveys..."

Line 146: "when observing permafrost"

Line 147: I know what you mean here, but this sentence needs to be rewritten. Please refer to the literature.

Figure 3: It is not clear to me when did you make the measurements? Same day as the photos?

Line 151: "Four-electrode"?

Line 152: What were the conditions?

Line 152: "conditions in the field"

Line 154: "over the moraine ridge"

Line 156: I feel like the information about the length of the profile should have come earlier in the section.

Line 158: "on an area without any ice and snow cover"

Line 161-163: Last sentence sounds bad in English. I know what you are trying to say though. Please refer to the literature and phrase this a bit better. For example: " In order to obtain a model of electrical resistivity an inversion algorithm was used in which a starting resistivity model is iteratively adjusted in order to achieve the best fit with the measured apparent resistivity values..... "

**3. Results and discussion**

**3.1**

Line 165: The title of this section is a bit misleading, ERT is also a type of geophysical investigation and is not presented here. I suggest renaming this section to refer strictly to GPR.

Line 167: "We acquired nine radargrams..."

Line 168: What other 2 profiles?

Line 170: "In Fig. 4"

Line 171: Same as before, saying it was "difficult" is redundant if it has not impacted the data in any way? Please expand.

Line 174: "The are covered by profiles..."

Line 175: "by snow during the previous year"

Line 180: "was used"

Line 187: A suggestion. Rather than listing them in the text, you can put the velocities in the stratigraphy table. By the way, I like your idea of having a summary table.

Line 189: " They have no scientific meaning and are the result of technical difficulties..."

Line 191: This sentence is a bit redundant. If I see the figure I know it contains those profiles, you can use this section to present the results and discuss them.

Line 192: "In order to capture the discontinuities"

Line 192: "between the start and end"

Line 193: Shouldn't this change in elevation be taken into consideration?

Line 193: "In both radargrams"

**3.2**

Line 215: Again, this title sounds misleading too. Isn't what you presented in the previous section also about the subsurface structure? Needs renaming.

Line 217: "Located in the lower part of the investigated area"

Line 217: "these horizontal profiles..."

Line 219: In reference to the topography data: 1. You should mention this in the methods section and 2.

Why can't you do this for all GPR sections? please explain.

Line 223: "the apparent resistivity values"

Line 224: how do you know those rock sizes?

Line 226: In reference to your claim that you are detecting marble. I don't think that is a valid assumption unless is supported with further evidence, that resistivity range can cover many types of material.

Line 230: These variation in size are very interesting, tied to the hydrodology of the area. It deserves a closer look.

Line 238: Try to discuss and expand this, it looks as if some melting occurred.

Line 245: Quite a redundant sentence again.

Line 250: You have said this before. The interesting bit here is that they match. Please discuss how well they match.

Line 253: Delete "right on the figure"

Line 254: "ERT data reaches a smaller depth"

Line 255: "with the area identified as a frozen zone on the GPR profiles"

Line 259: Delete "even in the sunniest days of summer". I don't know if that was the sunniest day that summer...please avoid vague sentences like this.

Line 263: "Snow patches were not observed"

Line 264-266:   I am not sure how these 2 sentences are relevant if you can't show any data or reference any study.

Line 271: This is an interesting point. You can expand.

Figure 10: It is not very clear to me exactly where do they overlap. Are the 2 tomograms at different scales?

Figure 11: The elements of this figure are way too small, I cannot see the labels at all. Every figure needs to be readable, even though you are reproducing some material you have shown before.

**4. Conclusion**

Line 278: "and the subsurface beneath it."

Line 280-283: I know the conclusion usually contains a bit of repetition, but you have almost copied the whole paragraph from the discussion. The conclusion should be a concise version of your results and discussion.

Line 285: "However, the thickness values estimated are a good base for monitoring microglaciers of this size."

Line 286: "The underlying layer is identified as a glacier bed ...and consists of rock blocks of different sizes, spaces between them being filled by ice, water and air."

Line 288: "beneath the surface as surface water is rarely observed."

Line 288: You did not discuss this conclusion in the discussion section. It is an interesting point that deserves consideration.

Line 290: "From the GPR profiles situated on the lower part of the investigated area..."

Line 292: "60000 $\Omega$.m, values typical for ice"

Line 299-302: Please remove these lines as they are not supported by any data or past studies.

---

## Author Comment (AC1)

Review of the manuscript entitled: GEOPHYSICAL MEASUREMENTS OF PERENNIAL SNOW PATCHES IN PIRIN MOUNTAIN, BULGARIA

General comments The research article enclosed aims to characterize perennial snow patches and the structure of their under- lying formations in Pirin Mountains, Bulgaria, using geophysical methods, such as: Ground Penetrating Radar and Electrical Resistivity Tomography. First of all, even though this type of characterization has been done before in other parts of the world, I believe it is novel and intriguing in this context, due to the rarity of microglaciers and perennial snows on the territory of Bulgaria, despite the high altitudes. I appreciate the hard work that has been put into this study. From personal experience, it is not easy to conduct geophysical surveys on mountainous terrain, especially if it is not easily accessible as this site appears to be. Normally, I would expect more repeatability of the surveys done throughout the years in order to capture the morphological/hydrological changes that might have occurred, but given the location, this may not have been an option. However, I do have several problems with this manuscript and in this current state I would only recom- mend a reconsideration with major revisions. I do believe this work is publishable, but needs a bit more work on the following aspects:

1. The use of English language. Even though I often understand the message behind a sentence, it is very hard to follow sometimes. There are a lot of grammatical errors and quite often I come across sentences that could be reworded or incorporated in a longer phrase to improve the readability of the paragraph. (I will give specific examples below and in some places I attempted to rephrase some parts in order for them to read better. I should say that I only picked up some mistakes, there are definitely others that would need attention).

**Thank you for your suggestions on improving the English language. We have modified the manuscript as suggested. Additionally, the language will be corrected by a colleague and a revised version will be uploaded when needed.**

2. I feel like there needs to be a bit more discussion about the changes you might have seen across the 3 years of surveys. All around us, nature is changing rapidly and cold regions are often the ones that show the greatest change. You touch a bit upon some structural changes you noticed on the tomograms imaging the microglacier, but I personally feel this should be expanded enriching the content of the manuscript.

3. There are a few technical issues that I came across when reading your methods and looking at the figures, but I think these can be overlooked if only they are properly explained (I will give details below).

**We are going to correct some of the figures**

I will give a list of specific comments below with line numbers associated. The ones with red headings I considered to be more pressing issues that need resolving. When I use quotation marks I am rephrasing that bit of the text or correcting a mistake.

Specific comments Abstract 1. Line 4: rephrase as "in Europe, situated in the...". This is an example of a sentence that could have been merged into a bigger phrase to improve readability.

**Done**

2. Line 5: ERT stands for Electrical Resistivity Tomography

**Done**

3. Line 6: "next three years" implies 2021 as well. Please rephrase.

**Done**

4. Line 11: No issue here, just wanted to underline this is an interesting find. Like I said above, the permafrost's changes in time would be interesting to look at. This could be a continuation of this work, a basis for an ampler future project.

**We are going to continue the measurements near Snezhnika but also near other perennial snow patches observed in high mountains of Bulgaria.**

1. Introduction

Line 22: "The most studied perennial snow"

**Done**

Line 25: "size were made"

**Done**

Line 25: What systematic measurements ? Be specific please. Also, "were conducted".

**Done**

Line 25: "at the end of summer"

**Done**

49 Line 31: There are many types of geophysical measurements, not just the ones listed here. A probe 50 reading a soil's temperature can be thought as a geophysical measurement. Are you sure none have been 51 carried out before? You could say: "Georgievna et al., (2019) finds that geophysical measurements, such 52 as... could be useful for a better site characterization because of their...".

Line 35: "Permafrost is also a good..."

**Done**

Line 45: Double parenthesis after Onaca et al.

**Done**

Line 52: " with the GPR technique used for imaging of..."

**Done**

Line 58: ERT is not an acronym for electrical resistivity. ER would be, if you want to use one.

**Done**

Line 63: I am not sure if you need the EM abbreviation if you don't use it afterwards.

**Done**

58 Line 79: ERT is not just in 2D, can be in 1,2,3 D and even timelapse 3D classed as 4D in some cases. 59 Please rephrase. Line 81: Do not forget salinity!

Line 93: The last sentence is a repetition, please rephrase or remove. In addition, the last paragraph is a bit too short, normally is meant to be setting the scene to what you are about to read in the paper. Please expand. For example you have not even mentioned which geophysical methods you have used, even though you are hinting that GPR and ERT are your methods of choice.

**Done**

2. Methods Line 97: You can just say ERT from now on. It was previously defined in the abstract and introduction and I suspect you will be asked for a list of abbreviations too.

**Done**

Line 98: I do not thing this intro paragraph is needed, at least not in this form. Everything you say here you repeat later on.

**Done**

2.1 71 Line 113: "as can be observed in Fig.2"

**Done**

Line 116: "in the same place over the microglacier's bed where the GPR profiles were made the previous year"

**Done**

Figure 2 caption: "over the years".

**Done**

75 Figure 2 caption: 2nd sentence should not be in a caption. 76 Figure 2: By

looking at the dates when the photos were taken, can you discuss a bit if the weather at the 77 specific time of year had an impact on your measurements? In 2017 and 2018 you went there in August, 78 whereas in 2019 and 2020 you went in October.

2.2 Line 133: "Locations are presented in..."

**Done**

Line 133: "Profile coordinates..."

**Done**

Line 134: 1-2 m seems like a rather big error to me. Can you justify that this is acceptable?

Line 135: The "relief's horizontals" ?

**Done**

Line 139: I really don't get why they have no topography, doesn't it skew the results? surely the 85 topography is not that similar across all radargrams. 86 Line 141: Even though I sympathise with the effort needed to perform this kind of survey, unless you 87 describe how this has impacted your results, this last sentence is unnecessary.

2.3 Line 144: "Based on other surveys..."

**Done**

Line 146: "when observing permafrost"

**Done**

91 Line 147: I know what you mean here, but this sentence needs to be rewritten. Please refer to the 92 literature. 93 Figure 3: It is not clear to me when did you make the measurements? Same day as the photos?

Line 151: "Four-electrode"?

**Done**

95 Line 152: What were the conditions?

Line 152: "conditions in the field"

**Done**

Line 154: "over the moraine ridge"

**Done**

98 Line 156: I feel like the information about the length of the profile should have come earlier in the section.

Line 158: "on an area without any ice and snow cover"

**Done**

100 Line 161-163: Last sentence sounds bad in English. I know what you are trying to say though. Please 101 refer to the literature and phrase this a bit better. For example: " In order to obtain a model of electrical 102 resistivity an inversion algorithm was used in which a starting resistivity model is iteratively adjusted in 103 order to achieve the best fit with the measured apparent resistivity values..... "

3. Results and discussion 3.1

106 Line 165: The title of this section is a bit misleading, ERT is also a type of geophysical investigation and 107 is not presented here. I suggest renaming this section to refer strictly to GPR.

Line 167: "We acquired nine radargrams..."

**Done**

109 Line 168: What other 2 profiles?

Line 170: "In Fig. 4"

**Done**

111 Line 171: Same as before, saying it was "difficult" is redundant if it has not impacted the data in any 112 way? Please expand. 113 Line 174: "The are covered by profiles..." 114 Line 175: "by snow during the previous year"

Line 180: "was used"

**Done**

Line 187: A suggestion. Rather than listing them in the text, you can put the velocities in the stratigraphy table. By the way, I like your idea of having a summary table.

**Thank you for this idea, we moved the GPR velocities in the stratigraphy table.**

Line 189: " They have no scientific meaning and are the result of technical difficulties..."

**Done**

Line 191: This sentence is a bit redundant. If I see the figure I know it contains those profiles, you can use this section to present the results and discuss them.

**Done**

Line 192: "In order to capture the discontinuities"

**Done**

Line 192: "between the start and end"

**Done**

123 Line 193: Shouldn't this change in elevation be taken into consideration?

Line 193: "In both radargrams"

**Done**

3.2

126 Line 215: Again, this title sounds misleading too. Isn't what you presented in the previous section also 127 about the subsurface structure? Needs renaming.

Line 217: "Located in the lower part of the investigated area"

**Done**

Line 217: "these horizontal profiles..."

**Done**

130 Line 219: In reference to the topography data: 1. You should mention this in the methods section and 2. 131 Why can't you do this for all GPR sections? please explain.

Line 223: "the apparent resistivity values"

**Done**

133 Line 224: how do you know those rock sizes? 134 Line 226: In reference to your claim that you are detecting marble. I don't think that is a valid assumption 135 unless is supported with further evidence, that resistivity range can cover many types of material. 136 Line 230: These variation in size are very interesting, tied to the hydrodology of the area. It deserves a 137 closer look. 138 Line 238: Try to discuss and expand this, it looks as if some melting occurred. 139 Line 245: Quite a redundant sentence again. 140 Line 250: You have said this before. The interesting bit here is that they match. Please discuss how well 141 they match.

Line 253: Delete "right on the figure"

**Done**

Line 254: "ERT data reaches a smaller depth"

**Done**

Line 255: "with the area identified as a frozen zone on the GPR profiles"

**Done**

Line 259: Delete "even in the sunniest days of summer". I don't know if that was the sunniest day that summer...please avoid vague sentences like this.

**Done**

Line 263: "Snow patches were not observed"

**Done**

148 Line 264-266: I am not sure how these 2 sentences are relevant if you can't show any data or reference 149 any study. Page 5150 Line 271: This is an interesting point. You can expand. 151 Figure 10: It is not very clear to me exactly where do they overlap. Are the 2 tomograms at different 152 scales? 153 Figure 11: The elements of this figure are way too small, I cannot see the labels at all. Every figure needs 154 to be readable, even though you are reproducing some material you have shown before.

4. Conclusion Line 278: "and the subsurface beneath it."

**Done**

157 Line 280-283: I know the conclusion usually contains a bit of repetition, but you have almost copied the 158 whole paragraph from the discussion. The conclusion should be a concise version of your results and 159 discussion.

Line 285: "However, the thickness values estimated are a good base for monitoring microglaciers of this size."

**Changed**

Line 286: "The underlying layer is identified as a glacier bed ...and consists of rock blocks of different sizes, spaces between them being filled by ice, water and air."

**Done**

Line 288: "beneath the surface as surface water is rarely observed."

**Done**

165 Line 288: You did not discuss this conclusion in the discussion section. It is an interesting point that 166 deserves consideration.

Line 290: "From the GPR profiles situated on the lower part of the investigated area..."

**Done**

Line 292: "60000 $\Omega.m$, values typical for ice"

**Done**

169 Line 299-302: Please remove these lines as they are not supported by any

data or past studies.

---

## Author Comment (AC2)

Reply to Anonymous Referee2

The study entitled 'Geophysical measurements of perennial snow patches in Pirin Mountain, Bulgaria, by Kisyov, A., Tzankov, C and Georgieva, G. presents interesting results in a region still poorly investigated so far. The paper brings valuable knowledge from a small glacier in the Pirin Mountains (Bulgaria), but should be highly improved to be published in this journal. The authors should address several important problems before the paper can be accepted. The paper is clearly structured and well-illustrated, but requires some supplementary explanations regarding the study site and the methodological approach. In addition, interpretations should be improved, new figures inserted and confusion regarding the inappropriate usage of some concepts should disappear. The English needs some smoothing in places.

After careful consideration, I recommend that the paper be published only after the authors address the issues listed below.

General comments

In several previous papers, the ice bodies assessed in this paper in the Pirin Mountains are called 'glaciarets' (Gachev et al., 2016) or 'microglaciers' (Grunewald et al., 2006) (even you mention in the paper Snezhnika as being a microglacier!). In addition, they are considered the southernmost glaciers in Europe by Hughes (2008), Grunewald and Scheithauer (2010) and other authors. Gachev (2017) mention typical glacial processes associated with these glaciarets, such as: striations and initiation of small moraines, suggesting that these ice bodies display motion and play a role in the present-day morphodynamics in the proglacial area. Moreover, the drillings performed by Grunewald in 2006 in Snezhnika revealed the presence of ice (Grunewald and Scheithauer, 2010). These glaciarets are probably several hundred years old (at least from the XIXth century). In this context, I think the authors should consider changing 'perennial snow patches' with 'glaciarets'.

**Thank you for the suggestion. We started the projects and later the paper with the idea to prove that the observed snow patches in high mountains of Bulgaria at the end of the summer are perennial. In the present version of the manuscript only the results for Snezhnika microglacier are presented. We will consider your suggestion and will replace "perennial snow patches" with "glaciarret" or "microglacier". We will add also some more explanation what is meant as "glacierret" because the word is not widely used in the literature.**

According to the title and objectives, the approach deals with geophysical measurements of perennial snow patches. However, the article would have a broader impact if the achieved results are used to gain knowledge regarding e.g., the evolution of these small glaciers, present-day changes/ behaviour of glaciaret, glacial-periglacial processes at this site, hydrological significance etc. In the present form, the article focused on identifying several different layers on GPR radargrams/ERT profiles, but the considerations regarding these layers' geomorphological/ hydrological importance are lacking almost completely. Therefore, I suggest going further with the analysis and interpretations than only identifying the bedrock depth, the permafrost, etc., but trying to explain the relevance of these findings for the mountain cryosphere. Otherwise, I am afraid that the paper seems to make an impact only locally.

**Thank you for the comment and suggestions. The manuscript was first submitted with a second part describing the measurements of another perennial snow patch in Banski Suhodol valley (a neighboring valley of Golyam Kazan cirque). The title has remained unchanged after we removed this part. We are going to think about a correction of the title. An extension of discussion part is also intended in which we will add more interpretation and analysis as suggested.**

I have some concerns regarding the design of the approach.

First, I didn't understand why the authors performed geophysical measurements on different alignments in different years? In the beginning, I thought that the authors would like to compare the results and quantify the changes, but it seems that was not the case. Because the profiles were not conducted on precisely the same lines, quantifications are not possible.
Second, the distribution of the profiles is not adequate. Most of the profiles were performed in the downslope part of the glacier, where the glaciaret is thin. It would have been good to have at least 1-2 transversal profiles in the upper part of the glaciaret (in this case, you could have calculated the glacier volume and then the water equivalent etc.).
Third, you used a simple handheld GPS, which can have low accuracy in this type of environment. Therefore, the profiles' exact position might be different from what appears on your map.

**1. All measurements were made within low budget projects for students with the main idea to demonstrate the capabilities of the available equipment for studying snow and ice patches as there are no big glaciers in Bulgaria. The interesting results and the very few of such studies were motivation for us to prepare a publication. It was not possible to make the measurements in the**

exact same date every year due to many reasons (the weather in the mountain for example) but also we don't think that it is so important in this case. The time of measurements was selected to be at the end of the summer and before the first snow in the mountain or the time of the expected lowest size of perennial snow patches. This is actually relative time, because in on year the first snow falls in September and in the other, the weather in October is still warm. Even the profiles are not conducted on the same lines an assesment of change within the years can be made.

2. The distribution of the profiles was meant to be more dense but the steep slope of the microglacier didn't allow us to make more profiles in the upper part in 2018. There is clarification about this in the text. We made more GPR profile later in 2020 in order to supplement the data about the thickness of Snezhnika and underlying structure.

3. Thank you for this comment. The accuracy of the handhels GPS is 2-3 m and this is less than the obtained subsurface structures. You are right that the quantification are not possible although some changes can be estimated. Actually the accuracy of the profiles positions was improved with the LUA images used later to produce DEM. This is not explained deeply in the text while DEM is not used for the calculations. We decided that including this information will encumber the text. Although we will add some more information in the manuscript. We were not able to use geodetic GPS in the previous studies but we will consider using it in the future as we are going to continue the monitoring of this site and also to conduct measurements in new places in the mountain.

You used ERT to investigate the lower part of the glaciaret, but as far as I know, ERT is problematic when the electrodes are fixed in the snow. Please refer to similar studies using ERT on snow patches/small glaciers and highlight the capabilities/limitations of ERT on snow/ice surface. In addition, I didn't understand why setting the electrodes distance at only 1.5 m? Because the distance between electrodes was small, the penetration was not enough to estimate 'frozen areas' thickness in some profiles. Generally, the measurement protocol in this environment uses a 5 m electrode distance.

Yes, the ERT is problematic when it is used fully on snow. There are laboratory experiments demonstrating the use of special electrodes in ice but it was not working in real measurements in the mountain. In our case all electrodes were in the gravel and only one electrode from one profile was in the ice. This is mentioned in the text.

The distance between electrodes was selected as the maximum possible length due to the terrain divided by 24 (the number of electrodes). Using a smaller distance between the electrodes we have better resolution. If we made measurements with lower resolution, we would miss the small anomalies like the watered layer near the surface. There is also clarification in the text about the distance between the electrodes.

It would also be helpful to mention the precise date of geophysical measurements each year.

Thank you for this suggestion, we will add this information in the text.

The radargrams have no topography, and because of this is difficult to interpret the reflections.

We have added topography according the similar comment of the Referee 1.

A recent study (Persoiu et al., 2021) showed that significant changes might occur at Snezhnika between different hydrological years (e.g., 2018 vs 2019). Therefore, please consider the interannual changes of this glaciaret when interpreting the results of profiles performed in different years. From the pictures, it seems that in 2018 was much more snow than in the following years. Do your GPR profiles tell you anything about ice-thickness changes between 2018 and 2020? Because according to Persoiu et al., significant variations in the surface may occur at this site.

The periodic changes in size of Snezhnika microglacier are observed since 1994 (Gachev et al.,2016). Between 2018 and 2020 we also observed the decrease of its size. The GPR profiles from 2018 show well the thickness of the "new" snow. The changes in size of the microglacier and how this affects the results from geophysical measurements are described in the text.

Because this site is unknown to most readers, you should give more details about this site. First, please include a map with the localization of the study area. Then, please add a short description of the evolution of glaciers in the Pleistocene and Holocene in this area supported by the morphology of this valley (e.g., moraines). Because karstic rocks occur here, please also refer to the presence of karstic features in this cirque. You did measurements in the proglacial area of the glacier, but you didn't describe it: type of surface, vegetation, clasts dimensions, presence of soil, water etc. It is also essential to describe the climate in the Pirin Mts.

**Thank you for this suggestion. We are going to prepare a more representative figure of the study site. We will include also information for evolution of glaciers in Pirin. According to the description of the site and the climate we can add a few sentences but mainly we think that the information given in 2.1 in the text is enough for this manuscript.**

Please refer to other similar studies regarding the interpretation of the GPR measurements. For example, you interpreted a pattern of reflections in the substrate as a' frozen zone'. Based on what characteristics of the reflections do you make this interpretation? Are there similar findings in other studies? The same observation for the second layer where the 'voids are filed with ice and water'. How do you be so sure that the voids between blocks are filled with ice and water? If the voids are filled with ice in this layer, then this is also permafrost. Then, what is the difference between layer 2 and 3? You should interpret all the other reflections by comparing them with similar findings elsewhere. Have you noticed any hyperbola in the radargams? You can use it to calculate velocities and see whether you have ice/permafrost/rocks or a mixture (most probably). What about internal coarse layers embedded in the ice? Grunewald and Scheithauer (2010) found such layers in the drillings done in 2006. Please also discuss the transition between snow/firn/ice.

**Interpretation of geophysical data (from one method) only by pattern is mainly ambiguous. In most papers there is previous information for the structure form boreholes or other methods. Due to the lack of previous information especially for the underlying structure of Shnezhnika we have compared the results from GPR and ERT measurements. The zone estimated as "frozen zone" on GPR profiles has very high resistivity on ERT. According to the small reflections within this zone on GPR profile it should be an ice rich zone. There are very few hyperbolas with well-preserved shapes. The estimated velocities range from 0.12 to 0.16. We used an average values for the different layers we distinguished. We outlined some internal layers within the ice on the radargrams from 2020. Thank you for the comment, we should extend discussion, adding more information and interpretation.**

One of the most interesting findings in the geophysical profiles is the so-called 'frozen zone'. Unfortunately, the interpretation based on the presented results is partly vague. For example, it is not clear if there is a lens of massive ice in the substrate or a mixture of ice and rocks (ice-cemented materials). The term 'frozen zone' is problematic and I suggest replacing it with ground ice/permafrost. First, you should clarify if you have periglacial or glacial ice in the substrate and discuss the origin of the ground ice/permafrost. Then you should describe the mechanisms involved in forming the permafrost at this site below the glacier and the non-frozen bedrock and whether it is ice-rich permafrost or massive ice is missing. Finally, try to explain processes that control permafrost occurrence at this site below an unfrozen/frozen ?? (this is not clear) bedrock and what happens with water below the glacier. Since this is a region with karstic rocks, please also refer to the hydrogeology in the Discussions and the presence/absence of caves/dolines etc. in the region.

**Thank you for this comment, it is very useful for us and we are going to extend the discussion part including the suggested information. Most probably, there is mixture of ice and rock blocks.**

Specific comments

Abstract

Line 7: "in order to evaluate changes in the snow patches size and thickness"... replace with "in order to assess glaciaret thickness and its internal structure". You haven't cuantified changes of size/thickness.

**Actually we made each year also measurements of the size of the microglacier but this information is not included in the text. We will consider your suggestion for correction.**

Line 8: Maximum thickness of ice can be higher than 8 m in the upper part of the glaciaret where there are no transversal profiles. Please add that maximum thickness of 8 m or even higher occur in the upper part of the glaciaret.

**We are going to mention that the thickness can be higher than estimated during our measurements.**

Line 10: replace "frozen zone" with permafrost/ice-cemmented sediments.

**Replaced**

Line 11: the presence of permafrost in the Pirin was also indicated by Onaca et al., (2020, 2022).

**Onaca et al. 2020 is cited in the Introduction part. Onaca et al. 2022 was submitted on 29 December 2021 when our manuscript was still available for discussions. It was not possible to refer a future work.**

Introduction

Please write a paragraph on the importance of knowing the ice thickness, internal structure for glaciology/hydrology/geomorph

**Thank you for this suggestion, we will add this information in the text.**

Line 18: add a citation after 'global changes than glacier'.

**Reference is added.**

Line 19-20: What do you want to say with "permafrost is the last stage of glacial life cycle"? This doesn't seem right. The occurrence of permafrost is not necessarily conditioned by the presence of a glacier. For example, in never-glaciated regions in Canada permafrost exists for several hundred of thousand years. In mid-latitude mountains, in regions without glaciers in the last 10 ka, permafrost still exists due to favourable topo-climatic conditions. In the Pirin Mts., permafrost probably also occurred at sites free of ice in the last 10 ka.

**Thank you for this comment. We will rewrite this sentence to be more clear.**

Line 21: This is wrong! During LIA the only glaciarets in Bulgaria were very small (see Gachev, 2000, Holocene glaciation in the mountains of Bulgaria, Mediterranean Geoscience Review, 2, 103-117). Large glaciers occurred in Bulgaria only in the Pleistocene. Please refer to this (see Kuhlemann et al., 2013, QI).

**Thank you for the comment. We will correct the sentence.**

Line 32: please see this recent study (Onaca et al., 2022) in which geophysical measurements on Snezhnika are presented.

**Thank you for the reference. It is released after the our manuscript was available for discussion.**

Line 36: it is not clear if you are talking about permafrost or air temperature?? Please also add a citation here.

**Thank you for the comment, we will correct the sentence.**

Line 36: not only "mountain slopes with permafrost are significantly vulnerable to climate change"; flat permafrost terrain is also vulnerable (see, Biskaborn et al., 2019).

**Yes, you are right, but in the Introduction we focus on mountain areas as the study is carried out in the mountain.**

Line 41: you are right that snow acts as a shield for radiation, but on the other hand it also may hamper the aggradation of permafrost.

**Thank you for the comment, we will add it in the text.**

Lines 74-75: "The polar ice...." - this is irrelevant here.

**Thank you for this comment, we will remove this part of the sentence.**

Line 84: "ERT can successfully be applied for studying glacial structures" - What types of structures? Please be explicit and add citations.

**We will refer to Kneisel et al. 2008 and wi will correct the sentence.**

Line 92: You didn't present any results from Banski Suhodol Valley and since is not the subject of this paper you should avoid referring to this site when presenting the aim of the paper.

**A correction of the text is made.**

Methods

Line 98: You didn't present any DEM in the paper. Please delete this sentence.

**We decided to leave this and to add some information for the DEM and the work with LUA on site.**

Line 99: 2.1. Study site description – please give more details on this site. A localization map + a detailed map of the topography of this cirque is also necessary. Please indicate on this map: Dzhamdziev ridge and all the other peaks.

**Thank you for this suggestion.**

Line 103: replace "snow patch" with "glaciaret".

**Done**

Line 112: "They were formed during the final phase...". It is not clear who?

**The sentence is rewritten.**

Line 131: What about mean velocities of snow? And permafrost? You mention that this glaciaret is a snow patch, but using the velocities for ice. In other studies ice is 0.16 m/ns. How can affect the thickness estimation of the glaciaret?

**The surface of Snezhnika in 2018 was presented with a wet snow layer, which was also compacted and semi-frozen, up to 2 m thick mainly at the lower part of the glacier, so we used the same average velocity 0.15m/ns as for the ice below. The change from 0.15 m/ns to 0.16 m/ns (+0.01 m/ns) will add 6.7 cm to every meter from the glacier section. This means that about 0.54 will by added to the estimated depth of the glacier in our deepest investigated part.**

Line 134: I have serious doubts regarding such a low error of the GPS in this shaded cirque. What about the vertical error? Topography is extremely important for the interpretation of geophysics profiles. When doing geophysics in such a rough terrain the protocol says that differential GPS is mandatory!

**The horizontal accuracy of the handhels GPS is 2-3 m and this is less than the obtained subsurface structures. The vertical error is double the horizontal. We were not able to use geodetic GPS in the previous studies but we will consider using it in the future. We are going to continue the monitoring of this site and also to conduct measurements in new places in the mountain.**

Lines 135-140: it is not clear if there are GPR profiles repeated exactly on the same line in 2020 compared with 2018. From fig 3 it seems that GPR in 2020 is different than those performed in 2018. It means that you can not actually compare the radargrams, by means of changes.

**We don't have such profiles. The effort was to cover more are of Snezhnika with GPR profiles in order to estimate better the thickness of the microglacier and its internal and underlying structure.**

Line 140: why didn't use topography when creating the radargrams? Topography is extremely important for interpretation. Without topography how can you interpret if reflections are parallel with the surface etc?

**The topography is added to the radargrams according to comments from Referee1.**

Line 142: It would have been good to try at least 1 or 2 GPR profiles in the upper part of the glaciaret in 2018 (when the glacieret size was the greatest in the last years) and where the thickness is probably greater.

**It was planed so but the slope was very steep and the antenna moved up and down along the slope, the people making the measurements walked faster or slower in different part of the profile which produced side effects on the radargrams. It is mentioned in the text but the problems during the measurements are not described in detail.**

Figure 3: Give more details about the picture in the background (when it was taken?). If possible, would be good to overlap the contour of glaciaret (or at least of the front) in 2018, 2019,2020 to see if it was ice in 2019 and 2020 where you did some profiles. Please replace Glacier Snezhnika with glaciaret Snezhnika on the picture. In the caption replace the Golyam Kazan area with Snezhnika glaciaret.

**We have new picture according to the comments of Referee 1.**

Line 155: Why setting the distance between electrodes at 1,5 m? Following the protocols in permafrost environments a distance of 5 m between electrodes allows you to measure 120 m profile length and probably around 20 m penetration depth. The moraine looks a bit challenging, but it would have been so interesting to make at least a profile on it, to see its internal structure!

**For the distance between the electrodes please see the answer above. The inner slopes of the moraine are very steep and this will cause false anomalies due to deformation of the profile geometry. Additionally the inner slope of the moraine is very unstable.**

Line 159: It is not clear if some profiles/parts of the profiles cross the glacieret. It seems that ERT 3 and 2 cross the glacieret and in this case, you should interpret the ERT values with extreme caution, since ERT in the snow is extremely tricky. Write a phrase about the contact between the electrodes and the ground?

**Thank you for this comment, we should explain it better in the text. Only one electrode from the last profile (the third) was in the ice. This was possible due to the shape of the microglacier and boarder between the ice and the gravel.**

Line 162: "real geoelectrical section" - what do you mean (inversion from apparent to true resistivity?)

**Yest it is inversion.**

Results and Discussion

Line 170: "which are horizontal relative to the slope"... how do you know, since your profile has no topography?

**This is corrected according to the comments of Referee 1.**

Line 170: replace "snowfield" with "glaciaret".

**Done**

Line 171: "The uppermost layer represents the microglacier". What do you mean? The ice?

**Yes, mainly ice. The first layer is the microglacier and other layers represent the structure beneath the microglacier.**

Less 173: you identified some discontinuities in the ice. Vey nice... can you say something about these?

**Thank you for this comment. This is mainly the discontinuity between the snow and the ice but also some discontinuities in the ice are notified. We will add a few sentences in the text to explain this.**

Lines 175-175: is not very clear here. Please rephrase.

**Thank you for the comment, we will correct this.**

Line 177: "The second layer lies under the ice"... you are not referring to GPR2018-1, GPR2018-2 and GPR2018-3, right?

**We are referring namely to these profiles. We will add also "snow" in the text.**

Line 178. You say that the voids are filled with water and ice in layer 2, but is not clear based on what you affirm this? Please, give references to similar findings. If this layer is draining the melted glacial water, why are some voids filled with icer? And how do you explain the presence of water in the so-called "meltwater zones", which are between the glacier front and the LIA moraine? Here is a possible scenario, but it might be wrong: the melting water may infiltrate layer 2, but because layer 3 is permafrost (impermeable), it follows the permafrost table downslope and accumulates in the proglacial area where ERT reveals a high concentration of water. Maybe if you agree with this scenario, you can make a simple model in which to represent the primary circuit of glacial melting water and the role of permafrost for drainage.

**We estimated the presence of meltwater zone from very law values of resistivity. We observed also on site near the microglacier water which disappeared very close to it. We think that the ERT profiles are not enough to make adequate model of the drainage system in the area but we will consider it during the future measurement.**

Line 186: This is very important. Can you comment on the large differences in the velocity between layers 2 and 3? Ice lenses mean massive ice (pure ice)? It is hard to believe... only if a thick mantle of debris-covered old glacial ice. I think here might be rather an ice-rich permafrost (usually around $105\Omega m$).

**The velocity of both layers is equalized later when we reproduced the radargrams according some comments of Referee1. Because of the lack of strong reflections we interpret this layer as very rich on ice permafrost. It is possible ice-rich glaciofluvial sediment. We are going to extend the discussion part with this.**

Line 194: replace "snowfield" with "glaciaret".

**Done**

Lines 199-200: It is not clear which figure is the frontal moraine? If it's 5a it means that the frontal moraine is well below the ice because GPR1 ends somewhere in the middle of the glacieret? Please clarify!

**Both profiles along the slope from 2020 end almost at lower border of Snezhnika. The size of the microglacier in 2018-2020 can be seen on fig.1.**

Line 210: according to Onaca et al., 2022 the maximum dept was 12 m. You should also refer to this finding.

**We will comment the estimated thickness from Onaca et al. 2022 in the Discussion.**

Line 222: apparent resistivity? Why not true resistivity?

**Thank you for the comment. We have remove the "apparent" from the text.**

Line 231-233: please be more precise: ground ice/permafrost (avoid 'frozen zone').

**Thank you for the comment, we will clarify the text according your suggestion.**

Line 238: Interesting finding! Can you explain why the active layer has thickened so much in only 1 year?

**In the first year of the study (2018) a snow layer was covering the most of the glacier bed area. In the summer there have had more melting (measurements were made in the end of August). While in the last year (2020) the snow layer was not presented, the size of the microglacier was smaller and also the measurements were made in the beginning of October. There was less melting.**

Line 256: How can you explain the occurrence of "frozen zones" only in the lowest part of the glaciaret?

**Thank you for the comment, we will consider explaining the reasons for it when we extend the discussion part. Several reasons can be given like for example the available data and penetration depth of the geophysical equipment. In the lower part of the microglacier (microglacier's bed) we can compare the results from GPR and ERT and based on both methods to estimate the presence of frozen zone (or ice rich permafrost). In the upper part we can compare only GPR but the profiles from 2020 are shallower.**

Lines 274-275: Not clear. Rephrase this.

**Thank you for the comment we will rephrase this part of the text.**

Please write a paragraph on the methodological uncertainties (mainly the limitations of GPR and ERT) and where the interpretations should be treated with cautions.

**Thank you for the suggestion, we will add this information.**

Conclusions

Line 278: replace "snow" with "ice".

**Done.**

Line 280: "...the ice body it reaches 8 m", but the maximum thickness may exceed this value in the upper part of the glacieret.

**8 m is the thickness according to our study. We explain in the text that we don't cover the whole area.**

Line 286 (and within all the manuscript): you are using rock blocks in many cases, but please refer to a classification of clasts based on the size of individuals (e.g.,pebbles, cobbles, boulders etc.).

**Thank you for this suggestion.**

Line 287: Please check again if ice occurs in layer 2.

**Zone 2 is the melted layer. The areas with bigger resistance are probably rocks and not permafrost**

Line 299: Indeed, shading is essential, but is not acting alone. You should consider the other controlling factors of permafrost in the Discussions.

**Thank you for the suggestion, we will consider it in extension of the discussion part.**

Line 300: I suggest to delete the last phrase. It is not a conclusion of this study.

**We will delete this sentence.**

References:

Biskaborn et al., 2019. https://doi.org/10.1038/s41467-018-08240-4

Gachev et al., 2020. DOI: 10.1007/s42990-020-00028-3

Kuhlemann et al., 2013. 10.1016/j.quaint.2012.06.027

Onaca et al., 2022. https://doi.org/10.1016/j.catena.2022.106143

Persoiu et al., 2021. https://doi.org/10.5194/tc-15-2383-2021 Citation: https://doi.org/10.5194/tc-2021-337-RC2

---

## Author Response (AR1)

Reply to Referee1, Mihai Cimpoiasu

General comments

The research article enclosed aims to characterize perennial snow patches and the structure of their underlying formations in Pirin Mountains, Bulgaria, using geophysical methods, such as: Ground Penetrating Radar and Electrical Resistivity Tomography. First of all, even though this type of characterization has been done before in other parts of the world, I believe it is novel and intriguing in this context, due to the rarity of microglaciers and perennial snows on the territory of Bulgaria, despite the high altitudes. I appreciate the hard work that has been put into this study. From personal experience, it is not easy to conduct geophysical surveys on mountainous terrain, especially if it is not easily accessible as this site appears to be. Normally, I would expect more repeatability of the surveys done throughout the years in order to capture the morphological/hydrological changes that might have occurred, but given the location, this may not have been an option. However, I do have several problems with this manuscript and in this current state I would only recommend a reconsideration with major revisions. I do believe this work is publishable, but needs a bit more work on the following aspects:

1. The use of English language. Even though I often understand the message behind a sentence, it is very hard to follow sometimes. There are a lot of grammatical errors and quite often I come across sentences that could be reworded or incorporated in a longer phrase to improve the readability of the paragraph. (I will give specific examples below and in some places I attempted to rephrase some parts in order for them to read better. I should say that I only picked up some mistakes, there are definitely others that would need attention).

**Thank you for your suggestions on improving the English language. We have modified the manuscript as suggested. Additionally, the language was corrected by a colleague who is proficient in English.**

2. I feel like there needs to be a bit more discussion about the changes you might have seen across the 3 years of surveys. All around us, nature is changing rapidly and cold regions are often the ones that show the greatest change. You touch a bit upon some structural changes you noticed on the tomograms imaging the microglacier, but I personally feel this should be expanded enriching the content of the manuscript.

**Your comments are very useful and the discussion has been extended.**

3. There are a few technical issues that I came across when reading your methods and looking at the figures, but I think these can be overlooked if only they are properly explained (I will give details below).

**The figures are corrected as suggested.**

I will give a list of specific comments below with line numbers associated. The ones with red headings I considered to be more pressing issues that need resolving. When I use quotation marks I am rephrasing that bit of the text or correcting a mistake.

Specific comments Abstract Line 4: rephrase as "in Europe, situated in the...". This is an example of a sentence that could have been merged into a bigger phrase to improve readability.

**Done**

Line 5: ERT stands for Electrical Resistivity Tomography

**Done**

Line 6: "next three years" implies 2021 as well. Please rephrase.

**Done**

Line 11: No issue here, just wanted to underline this is an interesting find. Like I said above, the permafrost's changes in time would be interesting to look at. This could be a continuation of this work, a basis for an ampler future project.

**We are going to continue the measurements near Snezhnika but also near other perennial snow patches observed in high mountains of Bulgaria.**

1. Introduction

Line 22: "The most studied perennial snow"

**Done**

Line 25: "size were made"

**Done**

Line 25: What systematic measurements ? Be specific please. Also, "were conducted".

**Done**

Line 25: "at the end of summer"

**Done**

Line 31: There are many types of geophysical measurements, not just the ones listed here. A probe reading a soil's temperature can be thought as a geophysical measurement. Are you sure none have been carried out before? You could say: "Georgievna et al., (2019) finds that geophysical measurements, such as... could be useful for a better site characterization because of their...".

**This sentence is rewritten.**

Line 35: "Permafrost is also a good..."

**Done**

Line 45: Double parenthesis after Onaca et al.

**Removed**

Line 52: " with the GPR technique used for imaging of..."

**Done**

Line 58: ERT is not an acronym for electrical resistivity. ER would be, if you want to use one.

**Done**

Line 63: I am not sure if you need the EM abbreviation if you don't use it afterwards.

**Done**

Line 79: ERT is not just in 2D, can be in 1,2,3 D and even timelapse 3D classed as 4D in some cases. Please rephrase.

**"2D Electrotomography"is removed.**

Line 81: Do not forget salinity!

**A correction with references is added in the text.**

Line 93: The last sentence is a repetition, please rephrase or remove. In addition, the last paragraph is a bit too short, normally is meant to be setting the scene to what you are about to read in the paper. Please expand. For example you have not even mentioned which geophysical methods you have used, even though you are hinting that GPR and ERT are your methods of choice.

**Done, we have removed the paragraph.**

2. Methods Line 97: You can just say ERT from now on. It was previously defined in the abstract and introduction and I suspect you will be asked for a list of abbreviations too.

**Done**

Line 98: I do not thing this intro paragraph is needed, at least not in this form. Everything you say here you repeat later on.

**Done**

2.1 71 Line 113: "as can be observed in Fig.2"

**Done**

Line 116: "in the same place over the microglacier's bed where the GPR profiles were made the previous year"

**Done**

Figure 2 caption: "over the years".

**Done**

Figure 2 caption: 2nd sentence should not be in a caption.

**Done**

Figure 2: By looking at the dates when the photos were taken, can you discuss a bit if the weather at the specific time of year had an impact on your measurements? In 2017 and 2018 you went there in August, whereas in 2019 and 2020 you went in October.

**The weather should not affect the results significantly, as we are investigating the thickness of the microglacier and the near subsurface structure. The size is more dependent on the time of measurements. According to some publications, the end of the hydrological cycle in the mountains of Bulgaria is in October and the measurements, especially of the size of perennial snow patches, should be made at the end of September or October. In some years, the first new snow in the mountains falls in September (like in 2017) and we decided to do all measurements at the end of August and to repeat them at the same time each year. The summers of 2019 and 2020 were very hot and we made measurements at the beginning of October, but the weather conditions were quite similar. Similar comments are added also in the text.**

2.2 Line 133: "Locations are presented in..."

**Done**

Line 133: "Profile coordinates..."

**Done**

Line 134: 1-2 m seems like a rather big error to me. Can you justify that this is acceptable?

**1-2 m is the error in horizontal coordinates according to the technical datasheet. We could not work with a geodetic instrument (like Trimble receivers) for the measurements in the projects. We think this errors is acceptable because the size of the microglacier is about 100x500 m and the profiles are about 8-10 m away from each other (the distance between the profiles was measured also with measuring tape onsite). The position of the profiles is also evaluated from the done images. Even with an error of 1-2 m the order of profiles will remain and thus, the description of the anomalies is correct. Within this error, we also assume that the GPR profiles (1,2,3 from 2018 and 3,4 from 2020) and ERT profiles overlap. The exact position is also checked from the**

**drone images. Comments are added in the text also.**

Line 135: The "relief's horizontals" ?

**Done**

Line 139: I really don't get why they have no topography, doesn't it skew the results? surely the topography is not that similar across all radargrams.

**All radargrams are reproduced with topography.**

Line 141: Even though I sympathise with the effort needed to perform this kind of survey, unless you describe how this has impacted your results, this last sentence is unnecessary.

**The sentence is removed.**

2.3 Line 144: "Based on other surveys..."

**Done**

Line 146: "when observing permafrost"

**Done**

Line 147: I know what you mean here, but this sentence needs to be rewritten. Please refer to the literature.

**Done**

Figure 3: It is not clear to me when did you make the measurements? Same day as the photos?

**The picture is made in 2018 in the day of measurements. This information is added also in the caption.**

Line 151: "Four-electrode"?

**Done**

Line 152: What were the conditions?

**The length of the profiles depends on the width of the glacier bed and save access to the moraines. Further information is added in the text.**

Line 152: "conditions in the field"

**Done**

Line 154: "over the moraine ridge"

**Done**

Line 156: I feel like the information about the length of the profile should have come earlier in the section.

**Done**

Line 158: "on an area without any ice and snow cover"

**Done**

Line 161-163: Last sentence sounds bad in English. I know what you are trying to say though. Please refer to the literature and phrase this a bit better. For example: " In order to obtain a model of electrical resistivity an

inversion algorithm was used in which a starting resistivity model is iteratively adjusted in order to achieve the best fit with the measured apparent resistivity values..... "

**Thank you for the suggestion, the sentence is corrected.**

3. Results and discussion 3.1 Line 165: The title of this section is a bit misleading, ERT is also a type of geophysical investigation and is not presented here. I suggest renaming this section to refer strictly to GPR.

**We decided to remove the subsections.**

Line 167: "We acquired nine radargrams..."

**Done**

Line 168: What other 2 profiles?

**The sentence is rewritten.**

Line 170: "In Fig. 4"

**Done**

Line 171: Same as before, saying it was "difficult" is redundant if it has not impacted the data in any way? Please expand.

**We decided to remove this sentence. The steep slope impacted more the measurements themselves and the security of the people during the measurements than the data. The impact on the data is explained later, so this information is not needed here.**

Line 174: "The are covered by profiles..."

**Line 174 does not contain such text. I don't understand this comment.**

Line 175: "by snow during the previous year"

**Done**

Line 180: "was used"

**Done**

Line 187: A suggestion. Rather than listing them in the text, you can put the velocities in the stratigraphy table. By the way, I like your idea of having a summary table.

**Thank you for this idea, we moved the GPR velocities in the stratigraphy table.**

Line 189: " They have no scientific meaning and are the result of technical difficulties..."

**Done**

Line 191: This sentence is a bit redundant. If I see the figure I know it contains those profiles, you can use this section to present the results and discuss them.

**Done**

Line 192: "In order to capture the discontinuities"

**Done**

Line 192: "between the start and end"

**Done**

Line 193: Shouldn't this change in elevation be taken into consideration?

**Yes, you are right. All radargrams are currently with topography and this sentence has been removed.**

Line 193: "In both radargrams"

**Done**

3.2 Line 215: Again, this title sounds misleading too. Isn't what you presented in the previous section also about the subsurface structure? Needs renaming.

**The title of subsection is removed.**

Line 217: "Located in the lower part of the investigated area"

**Done**

Line 217: "these horizontal profiles..."

**Done**

Line 219: In reference to the topography data: 1. You should mention this in the methods section and 2. Why can't you do this for all GPR sections? please explain.

**1. There is information about the DEM in the Methods section.**

**2. Explained in reply to line 139 and line 193.**

Line 223: "the apparent resistivity values"

**Done**

Line 224: how do you know those rock sizes?

**The rock sizes were estimated on-site during the measurements. The change in the size of the microglacier each year allows us to see the layer beneath the microglacier. This sentence is also modified in the revised version.**

134 Line 226: In reference to your claim that you are detecting marble. I don't think that is a valid assumption unless is supported with further evidence, that resistivity range can cover many types of material.

**The site where the microglacier is situated and generally this part of the Pirin Mountains is known as the marble ridge of Pirin. It is mentioned in subsection 2.1 that the rocks in the area are marble. Our claim is based mainly on this and the obtained resistivities are also in agreement.**

Line 230: These variation in size are very interesting, tied to the hydrology of the area. It deserves a closer look. Line 238: Try to discuss and expand this, it looks as if some melting occurred.

**Thank you for both suggestions. We have extended the discussion adding more comments on these changes.**

Line 245: Quite a redundant sentence again.

**The sentence is removed.**

Line 250: You have said this before. The interesting bit here is that they match. Please discuss how well they match.

**Thank you for this comment, we have changed the paragraph according you suggestions. We have also decided to removed one of the figures.**

Line 253: Delete "right on the figure"

**Done**

Line 254: "ERT data reaches a smaller depth"

**Done**

Line 255: "with the area identified as a frozen zone on the GPR profiles"

**Done**

Line 259: Delete "even in the sunniest days of summer". I don't know if that was the sunniest day that summer...please avoid vague sentences like this.

**Done**

Line 263: "Snow patches were not observed"

**Done**

Line 264-266: I am not sure how these 2 sentences are relevant if you can't show any data or reference any study. Line 299-302: Please remove these lines as they are not supported by any data or past studies.

**The sentences sugested in the both comments are removed from discussion and conclusions.**

Line 271: This is an interesting point. You can expand.

**Thank you for this comment, the discussion is expanded.**

Figure 10: It is not very clear to me exactly where do they overlap. Are the 2 tomograms at different scales? Figure 11: The elements of this figure are way too small, I cannot see the labels at all. Every figure needs to be readable, even though you are reproducing some material you have shown before.

**Both figures are corrected in the revised version.**

4. Conclusion Line 278: "and the subsurface beneath it."

**Done**

Line 280-283: I know the conclusion usually contains a bit of repetition, but you have almost copied the whole paragraph from the discussion. The conclusion should be a concise version of your results and discussion.

**Thank you for your suggestion, we have modified the Conclusion section accordingly.**

Line 285: "However, the thickness values estimated are a good base for monitoring microglaciers of this size."

**Changed**

Line 286: "The underlying layer is identified as a glacier bed ...and consists of rock blocks of different sizes, spaces between them being filled by ice, water and air."

**Done**

Line 288: "beneath the surface as surface water is rarely observed."

**Done**

Line 288: You did not discuss this conclusion in the discussion section. It is an interesting point that deserves consideration.

**We agree with the reviewer and are grateful for this useful comment. We have edited the discussions accordingly.**

Line 290: "From the GPR profiles situated on the lower part of the investigated area..."

**Done**

Line 292: "60000 $\Omega.m$, values typical for ice"

**Done**

Reply to Anonymous Referee2

The study entitled 'Geophysical measurements of perennial snow patches in Pirin Mountain, Bulgaria, by Kisyov, A., Tzankov, C and Georgieva, G. presents interesting results in a region still poorly investigated so far. The paper brings valuable knowledge from a small glacier in the Pirin Mountains (Bulgaria), but should be highly improved to be published in this journal. The authors should address several important problems before the paper can be accepted. The paper is clearly structured and well-illustrated, but requires some supplementary explanations regarding the study site and the methodological approach. In addition, interpretations should be improved, new figures inserted and confusion regarding the inappropriate usage of some concepts should disappear. The English needs some smoothing in places.

After careful consideration, I recommend that the paper be published only after the authors address the issues listed below.

General comments

In several previous papers, the ice bodies assessed in this paper in the Pirin Mountains are called 'glaciarets' (Gachev et al., 2016) or 'microglaciers' (Grunewald et al., 2006) (even you mention in the paper Snezhnika as being a microglacier!). In addition, they are considered the southernmost glaciers in Europe by Hughes (2008), Grunewald and Scheithauer (2010) and other authors. Gachev (2017) mention typical glacial processes associated with these glaciarets, such as: striations and initiation of small moraines, suggesting that these ice bodies display motion and play a role in the present-day morphodynamics in the proglacial area. Moreover, the drillings performed by Grunewald in 2006 in Snezhnika revealed the presence of ice (Grunewald and Scheithauer, 2010). These glaciarets are probably several hundred years old (at least from the XIXth century). In this context, I think the authors should consider changing 'perennial snow patches' with 'glaciarets'.

**Thank you for the suggestion. We started the projects and later the paper with the idea to prove that the observed snow patches in the high mountains of Bulgaria at the end of the summer are perennial. In the present version of the manuscript, only the results for the Snezhnika microglacier are presented. We considered your suggestion and replaced "perennial snow patches" with "glaciarret" or "microglacier". We have added also some more explanations of what is meant by "glacierret" because the word is not widely used in the literature.**

According to the title and objectives, the approach deals with geophysical measurements of perennial snow patches. However, the article would have a broader impact if the achieved results are used to gain knowledge regarding e.g., the evolution of these small glaciers, present-day changes/ behaviour of glaciaret, glacial-periglacial processes at this site, hydrological significance etc. In the present form, the article focused on identifying several different layers on GPR radargrams/ERT profiles, but the considerations regarding these layers' geomorphological/ hydrological importance are lacking almost completely. Therefore, I suggest going further with the analysis and interpretations than only identifying the bedrock depth, the permafrost, etc., but trying to explain the relevance of these findings for the mountain cryosphere. Otherwise, I am afraid that the paper seems to make an impact only locally.

**We have changed the title and we have extended the discussion part with more interpretation and analysis as suggested.**

I have some concerns regarding the design of the approach.

First, I didn't understand why the authors performed geophysical measurements on different alignments in different years? In the beginning, I thought that the authors would like to compare the results and quantify the changes, but it seems that was not the case. Because the profiles were not conducted on precisely the same lines, quantifications are not possible.
Second, the distribution of the profiles is not adequate. Most of the profiles were performed in the downslope part of the glacier, where the glaciaret is thin. It would have been good to have at least 1-2 transversal profiles in the upper part of the glaciaret (in this case, you could have calculated the glacier volume and then the water equivalent etc.).
Third, you used a simple handheld GPS, which can have low accuracy in this type of environment. Therefore, the profiles' exact position might be different from what appears on your map.

**1. All measurements were made within low-budget projects for students with the main idea to demonstrate the capabilities of the available equipment for studying snow and ice patches as there are no big glaciers in Bulgaria. The interesting results and the very few such studies were a motivation for us to prepare a publication. It was not possible to make the measurements on the same date every year due to many reasons (the weather in the mountain for example) but also we don't think that it is so important in this case. The time of measurements was selected to be at the end of the summer and before the first snow in the mountain or the time of the expected**

lowest size of perennial snow patches. This is actually a relative time because in one year the first snow falls in September and in the other, the weather in October is still warm. Even if the profiles are not conducted on the same lines an assessment of change within the years can be made.

**2. The distribution of the profiles was meant to be more dense, but the steep slope of the microglacier didn't allow us to make more profiles in the upper part in 2018. There is clarification about this in the text. We made more GPR profile later in 2020 in order to supplement the data about the thickness of Snezhnika and underlying structure.**

**3. Thank you for this comment. The accuracy of the handheld GPS is 2-3 m and this is less than the obtained subsurface structures. You are right that quantification is not possible although some changes can be estimated. Actually, the accuracy of the positions of the profile was improved with the UAV images used later to produce DEM. We have added more information to the text. We were not able to use geodetic GPS in the previous studies but we will consider using it in the future as we are going to continue the monitoring of this site and also conduct measurements in new places in the mountain.**

You used ERT to investigate the lower part of the glaciaret, but as far as I know, ERT is problematic when the electrodes are fixed in the snow. Please refer to similar studies using ERT on snow patches/small glaciers and highlight the capabilities/limitations of ERT on snow/ice surface. In addition, I didn't understand why setting the electrodes distance at only 1.5 m? Because the distance between electrodes was small, the penetration was not enough to estimate 'frozen areas' thickness in some profiles. Generally, the measurement protocol in this environment uses a 5 m electrode distance.

**Yes, the ERT is problematic when it is used fully on snow. Laboratory experiments are demonstrating the use of special electrodes in ice, but it was not working in on site measurements in the mountain. In our case, all electrodes were in the gravel, and only one electrode from one profile was in the ice. This is mentioned in the text.**

**The distance between electrodes was selected as the maximum possible length due to the terrain divided by 24 (the number of electrodes). Using a smaller distance between the electrodes, we have better resolution. If we made measurements with lower resolution, we would miss the small anomalies like the watered layer near the surface. There is also clarification in the text about the distance between the electrodes.**

It would also be helpful to mention the precise date of geophysical measurements each year.

**Thank you for this suggestion, the dates of the measurements are added now in the text and in the caption of Figure 2.**

The radargrams have no topography, and because of this is difficult to interpret the reflections.

**All figures including radargrams are reproduced with topography.**

A recent study (Persoiu et al., 2021) showed that significant changes might occur at Snezhnika between different hydrological years (e.g., 2018 vs 2019). Therefore, please consider the interannual changes of this glaciaret when interpreting the results of profiles performed in different years. From the pictures, it seems that in 2018 was much more snow than in the following years. Do your GPR profiles tell you anything about ice-thickness changes between 2018 and 2020? Because according to Persoiu et al., significant variations in the surface may occur at this site.

**The periodic changes in the size of the Snezhnika microglacier have been observed since 1994 (Gachev et al.,2016). Between 2018 and 2020, we also have observed a decrease in its size. The GPR profiles from 2018 show the thickness of the "new" snow. The changes in the size of the microglacier and how this affects the results from geophysical measurements are described in the text.**

Because this site is unknown to most readers, you should give more details about this site. First, please include a map with the localization of the study area. Then, please add a short description of the evolution of glaciers in the Pleistocene and Holocene in this area supported by the morphology of this valley (e.g., moraines). Because karstic rocks occur here, please also refer to the presence of karstic features in this cirque. You did measurements in the proglacial area of the glacier, but you didn't describe it: type of surface, vegetation, clasts dimensions, presence of soil, water etc. It is also essential to describe the climate in the Pirin Mts.

**Thank you for this suggestion. Figure 1 is changed, and the picture is replaced with a map of the area. We have added more information about the vegetation, the lack of surface water,**

**and the climate of the site. We decided to not describe the evolution of the glaciers in Pirin Mountain because it would extend the introduction part too much. We have added references in "Introduction" and "Study site description" to works, where this evolution is well described.**

Please refer to other similar studies regarding the interpretation of the GPR measurements. For example, you interpreted a pattern of reflections in the substrate as a' frozen zone'. Based on what characteristics of the reflections do you make this interpretation? Are there similar findings in other studies? The same observation for the second layer where the 'voids are filed with ice and water'. How do you be so sure that the voids between blocks are filled with ice and water? If the voids are filled with ice in this layer, then this is also permafrost. Then, what is the difference between layer 2 and 3? You should interpret all the other reflections by comparing them with similar findings elsewhere. Have you noticed any hyperbola in the radargams? You can use it to calculate velocities and see whether you have ice/permafrost/rocks or a mixture (most probably). What about internal coarse layers embedded in the ice? Grunewald and Scheithauer (2010) found such layers in the drillings done in 2006. Please also discuss the transition between snow/firn/ice.

**Interpretation of geophysical data (from one method) only by the pattern is mainly ambiguous. In most papers, there is previous information for the structure from boreholes or other methods. Due to the lack of previous information especially for the underlying structure of Shnezhnika, we have compared the results from GPR and ERT measurements. The zone estimated as "frozen zone" on GPR profiles has very high resistivity on ERT. According to the small reflections within this zone on the GPR profile, it should be an ice-rich zone. There are very few hyperbolas with well-preserved shapes. The estimated velocities range from 0.12 to 0.16. We used average values for the different layers we distinguished. We outlined some internal layers within the ice on the radargrams from 2020. Thank you for the comment, discussion part is extended with more interpretation, and reference is added for intepretation og GPR data.**

One of the most interesting findings in the geophysical profiles is the so-called 'frozen zone'. Unfortunately, the interpretation based on the presented results is partly vague. For example, it is not clear if there is a lens of massive ice in the substrate or a mixture of ice and rocks (ice-cemented materials). The term 'frozen zone' is problematic and I suggest replacing it with ground ice/permafrost. First, you should clarify if you have periglacial or glacial ice in the substrate and discuss the origin of the ground ice/permafrost. Then you should describe the mechanisms involved in forming the permafrost at this site below the glacier and the non-frozen bedrock and whether it is ice-rich permafrost or massive ice is missing. Finally, try to explain processes that control permafrost occurrence at this site below an unfrozen/frozen ?? (this is not clear) bedrock and what happens with water below the glacier. Since this is a region with karstic rocks, please also refer to the hydrogeology in the Discussions and the presence/absence of caves/dolines etc. in the region.

**Thank you for this comment, it is very useful for us. We have replaced most of the "frozen zone" with "ice rich permafrost" or "permafrost". We have added comments in the discussion on hydrology and the probable origin of the frozen zone.**

Specific comments

Abstract

Line 7: "in order to evaluate changes in the snow patches size and thickness"... replace with "in order to assess glaciaret thickness and its internal structure". You haven't cuantified changes of size/thickness.

**Actually we made each year also measurements of the size of the microglacier but this information was not included in the text. Now the text is corrected.**

Line 8: Maximum thickness of ice can be higher than 8 m in the upper part of the glaciaret where there are no transversal profiles. Please add that maximum thickness of 8 m or even higher occur in the upper part of the glaciaret.

**We have added "8 m or probably higher in some areas" in the text.**

Line 10: replace "frozen zone" with permafrost/ice-cemmented sediments.

**Replaced**

Line 11: the presence of permafrost in the Pirin was also indicated by Onaca et al., (2020, 2022).

**Onaca et al. 2020 is cited in the Introduction part. Onaca et al. 2022 was submitted on 29 December 2021 when our manuscript was still available for discussions. It was not possible to refer a future work, but it is added as reference in the revised version.**

Introduction

Please write a paragraph on the importance of knowing the ice thickness, internal structure for glaciology/hydrology/geomorph

**Thank you for this suggestion. Several sentences are added at the end of the introduction part.**

Line 18: add a citation after 'global changes than glacier'.

**Reference is added.**

Line 19-20: What do you want to say with "permafrost is the last stage of glacial life cycle"? This doesn't seem right. The occurrence of permafrost is not necessarily conditioned by the presence of a glacier. For example, in never-glaciated regions in Canada permafrost exists for several hundred of thousand years. In mid-latitude mountains, in regions without glaciers in the last 10 ka, permafrost still exists due to favourable topo-climatic conditions. In the Pirin Mts., permafrost probably also occurred at sites free of ice in the last 10 ka.

**Thank you for this comment. Here, it was meant that the permafrost occure near glaciers and perennial snow patches, but you are right, and we have rewritten the sentence.**

Line 21: This is wrong! During LIA the only glaciarets in Bulgaria were very small (see Gachev, 2000, Holocene glaciation in the mountains of Bulgaria, Mediterranean Geoscience Review, 2, 103-117). Large glaciers occurred in Bulgaria only in the Pleistocene. Please refer to this (see Kuhlemann et al., 2013, QI).

**Thank you for the comment. The sentence is corrected and a reference is added.**

Line 32: please see this recent study (Onaca et al., 2022) in which geophysical measurements on Snezhnika are presented.

**Thank you for the reference. It is released after the our manuscript was available for discussion.**

Line 36: it is not clear if you are talking about permafrost or air temperature?? Please also add a citation here.

**Thank you for the comment, the sentence is corrected and reference is added.**

Line 36: not only "mountain slopes with permafrost are significantly vulnerable to climate change"; flat permafrost terrain is also vulnerable (see, Biskaborn et al., 2019).

**Yes, you are right, but in the Introduction we focus on mountain areas as the study is carried out in the mountain.**

Line 41: you are right that snow acts as a shield for radiation, but on the other hand it also may hamper the aggradation of permafrost.

**Thank you for the comment, it is added in the text.**

Lines 74-75: "The polar ice...." - this is irrelevant here.

**Thank you for this comment, this part of the sentence is removed.**

Line 84: "ERT can successfully be applied for studying glacial structures" - What types of structures? Please be explicit and add citations.

**We refered to Kneisel et al. 2008. The sentence is corrected.**

Line 92: You didn't present any results from Banski Suhodol Valley and since is not the subject of this paper you should avoid referring to this site when presenting the aim of the paper.

**A correction of the text is made.**

Methods

Line 98: You didn't present any DEM in the paper. Please delete this sentence.

**Information for DEM is added in the revised version.**

Line 99: 2.1. Study site description – please give more details on this site. A localization map + a detailed map of the topography of this cirque is also necessary. Please indicate on this map: Dzhamdziev ridge and all the other peaks.

**Thank you for this suggestion, a map is added as Figure 1.**

Line 103: replace "snow patch" with "glaciaret".

**Done**

Line 112: "They were formed during the final phase...". It is not clear who?

**The sentence is rewritten.**

Line 131: What about mean velocities of snow? And permafrost? You mention that this glaciaret is a snow patch, but using the velocities for ice. In other studies ice is 0.16 m/ns. How can affect the thickness estimation of the glaciaret?

**The surface of Snezhnika in 2018 was presented with a wet snow layer, which was also compacted and semi-frozen, up to 2 m thick, mainly at the lower part of the glacier. Be used the same average velocity of 0.15m/ns for this layer as for the ice below. The change from 0.15 m/ns to 0.16 m/ns (+0.01 m/ns) will add 6.7 cm to every meter from the glacier section. This means that if we change velocity, about 0.54 m will be added to the estimated depth of the glacier in our deepest investigated part.**

Line 134: I have serious doubts regarding such a low error of the GPS in this shaded cirque. What about the vertical error? Topography is extremely important for the interpretation of geophysics profiles. When doing geophysics in such a rough terrain the protocol says that differential GPS is mandatory!

**The horizontal accuracy of the handheld GPS is 2-3 m, and this is less than the obtained subsurface structures. The vertical error is double the horizontal. The exact position of the profiles is also checked on the drone images. We were not able to use geodetic GPS in the previous studies, but we will consider using it in the future. We are going to continue the monitoring of this site and also conduct measurements in new places in the mountain.**

Lines 135-140: it is not clear if there are GPR profiles repeated exactly on the same line in 2020 compared with 2018. From fig 3 it seems that GPR in 2020 is different than those performed in 2018. It means that you can not actually compare the radargrams, by means of changes.

**We don't have such profiles. The effort was to cover more area of Snezhnika with GPR profiles to estimate better the thickness of the microglacier and its internal and underlying structure.**

Line 140: why didn't use topography when creating the radargrams? Topography is extremely important for interpretation. Without topography how can you interpret if reflections are parallel with the surface etc?

**The topography is added to the radargrams and all figures presenting GPR data are reproduced.**

Line 142: It would have been good to try at least 1 or 2 GPR profiles in the upper part of the glaciaret in 2018 (when the glacieret size was the greatest in the last years) and where the thickness is probably greater.

**It was planned so, but the slope was very steep, and the antenna moved up and down along the slope. The people making the measurements walked faster or slower in a different part of the profile, which produced side effects on the radargrams. It is mentioned in the text, but the problems during the measurements are not described in detail.**

Figure 3: Give more details about the picture in the background (when it was taken?). If possible, would be good to overlap the contour of glaciaret (or at least of the front) in 2018, 2019,2020 to see if it was ice in 2019 and 2020 where you did some profiles. Please replace Glacier Snezhnika with glaciaret Snezhnika on the picture. In the caption replace the Golyam Kazan area with Snezhnika glaciaret.

**The figure is changed and the suggested information is added in the caption.**

Line 155: Why setting the distance between electrodes at 1,5 m? Following the protocols in permafrost environments a distance of 5 m between electrodes allows you to measure 120 m profile length and probably around 20 m penetration depth. The moraine looks a bit challenging, but it would have been so interesting to make at least a profile on it, to see its internal structure!

**For the distance between the electrodes, please see the answer above. The inner slopes of the moraine are very steep, and this will cause false anomalies due to the deformation of the profile geometry. Additionally, the inner slope of the moraine is very unstable.**

Line 159: It is not clear if some profiles/parts of the profiles cross the glacieret. It seems that ERT 3 and 2 cross the glacieret and in this case, you should interpret the ERT values with extreme caution, since ERT in the snow is extremely tricky. Write a phrase about the contact between the electrodes and the ground?

**Thank you for this comment, we have explained it more in the text. Only one electrode from the last profile (the third) was in the ice. The measurement so was possible due to the shape of the microglacier and probably the thin debris layer above the ice in this part.**

Line 162: "real geoelectrical section" - what do you mean (inversion from apparent to true resistivity?)

**Yes, it is inversion.**

Results and Discussion

Line 170: "which are horizontal relative to the slope"... how do you know, since your profile has no topography?

**The figures are corrected.**

Line 170: replace "snowfield" with "glaciaret".

**Done**

Line 171: "The uppermost layer represents the microglacier". What do you mean? The ice?

**Yes, mainly ice and snow. The first layer is the microglacier and other layers represent the structure beneath the microglacier.**

Less 173: you identified some discontinuities in the ice. Vey nice... can you say something about these?

**Thank you for this comment. This is mainly the discontinuity between the snow and the ice but also some discontinuities in the ice are notified. A few sentences are added in the text to explain this.**

Lines 175-175: is not very clear here. Please rephrase.

**Done**

Line 177: "The second layer lies under the ice"... you are not referring to GPR2018-1, GPR2018-2 and GPR2018-3, right?

**We are referring namely to these profiles.**

Line 178. You say that the voids are filled with water and ice in layer 2, but is not clear based on what you affirm this? Please, give references to similar findings. If this layer is draining the melted glacial water, why are some voids filled with icer? And how do you explain the presence of water in the so-called "meltwater zones", which are between the glacier front and the LIA moraine? Here is a possible scenario, but it might be wrong: the melting water may infiltrate layer 2, but because layer 3 is permafrost (impermeable), it follows the permafrost table downslope and accumulates in the proglacial area where ERT reveals a high concentration of water. Maybe if you agree with this scenario, you can make a simple model in which to represent the primary circuit of glacial melting water and the role of permafrost for drainage.

**We estimated the presence of meltwater zone from very law values of resistivity. We observed also on site near the microglacier water which disappeared very close to it. We think that the ERT profiles are not enough to make adequate model of the drainage system in the area but we will consider it during the future measurement. More explanation is added in the text.**

Line 186: This is very important. Can you comment on the large differences in the velocity between layers 2 and 3? Ice lenses mean massive ice (pure ice)? It is hard to believe... only if a thick mantle of debris-covered old glacial ice. I think here might be rather an ice-rich permafrost (usually around $105\Omega m$).

**The velocity of both layers is equalized later when we reproduced the radargrams. Because of the lack of strong reflections we interpret this layer as very rich on ice permafrost. It is possible to be ice-rich glaciofluvial sediment. The discussion part is extended with comments on this.**

Line 194: replace "snowfield" with "glaciaret".

**Done**

Lines 199-200: It is not clear which figure is the frontal moraine? If it's 5a it means that the frontal moraine is well below the ice because GPR1 ends somewhere in the middle of the glacieret? Please clarify!

**Both profiles along the slope from 2020 end almost at lower border of Snezhnika. The size of the microglacier in 2018-2020 can be seen on fig.3 in the revised version.**

Line 210: according to Onaca et al., 2022 the maximum dept was 12 m. You should also refer to this finding.

**We commented the estimated thickness from Onaca et al. 2022 in the Discussion.**

Line 222: apparent resistivity? Why not true resistivity?

**Thank you for the comment. We have remove the "apparent" from the text.**

Line 231-233: please be more precise: ground ice/permafrost (avoid 'frozen zone').

**Thank you for the comment, we replaced "frozen zone" with permafrost.**

Line 238: Interesting finding! Can you explain why the active layer has thickened so much in only 1 year?

**We added more comments to the discussion. We assume that the weather and the shift of the profiles are reasons for the bigger change.**

Line 256: How can you explain the occurrence of "frozen zones" only in the lowest part of the glaciaret?

**Thank you for the comment. Mainly we don't have information so deep to answer this question. The GPR profiles from the upper part (from 2020) reach almost only the border between ice and the underlying layer. And on the upper profiles from 2018, pale rectangular areas are presented which makes the interpretation difficult. In the lower part of the microglacier (microglacier's bed), we can compare the results from GPR and ERT, and based on both methods to estimate the presence of frozen zone (or ice rich permafrost).**

Lines 274-275: Not clear. Rephrase this.

**Done.**

Please write a paragraph on the methodological uncertainties (mainly the limitations of GPR and ERT) and where the interpretations should be treated with cautions.

**Thank you for the suggestion, a paragraph including suggested information is added.**

Conclusions

Line 278: replace "snow" with "ice".

**Done.**

Line 280: "...the ice body it reaches 8 m", but the maximum thickness may exceed this value in the upper part of the glacieret.

**8 m is the thickness according to our study. We explain in the text that we don't cover the whole area, and it is possible to have greater depth.**

Line 286 (and within all the manuscript): you are using rock blocks in many cases, but please refer to a classification of clasts based on the size of individuals (e.g.,pebbles, cobbles, boulders etc.).

**Thank you for this suggestion.**

Line 287: Please check again if ice occurs in layer 2.

**Zone 2 is the melte water layer. The areas with bigger resistance are probably rocks and not permafrost**

Line 299: Indeed, shading is essential, but is not acting alone. You should consider the other controlling factors of permafrost in the Discussions.

**Thank you for the suggestion, it is commented in the discussion part.**

Line 300: I suggest to delete the last phrase. It is not a conclusion of this study.

**Done.**

References:

Biskaborn et al., 2019. https://doi.org/10.1038/s41467-018-08240-4

Gachev et al., 2020. DOI: 10.1007/s42990-020-00028-3

Kuhlemann et al., 2013. 10.1016/j.quaint.2012.06.027

Onaca et al., 2022. https://doi.org/10.1016/j.catena.2022.106143

Persoiu et al., 2021. https://doi.org/10.5194/tc-15-2383-2021 Citation: https://doi.org/10.5194/tc-2021-337-RC2

---

## Referee Report (RR1)

**General comments**

This is my second review of this manuscript, now entitled: "Geophysical measurements of the Southernmost microglacier in Europe suggest permafrost occurrence in Bulgaria".

I acknowledge the clear improvements done to the article. I can safely say it is a valuable piece of science and it will be a good contribution to the literature. However, still needs a bit more work before it can be published. At the moment, I cannot recommend it for publication.

First of all, the language issue still persists especially in the paragraphs where new text was added. There are still quite a lot of mistakes and several sentences difficult to understand. I have tried to help the authors with some suggestions as much as I could, but the whole text needs to be thoroughly examined and its language more rigorously improved.

There is a serious lack of detail about how ERT measurements were collected. You don't even mention the device used. How many measurements per profile? Did you acquire any reciprocal data? Any data filtering applied pre-inversion? Any error models applied to the data? Any temperature corrections? It looks like you used Res2DInv as an inversion software, but you are not referencing it. Why is that?

Finally, it would be useful to discuss the wider implications of your work. Why are your results relevant to the wider scientific community? Examples of a few discussion topics you might want to touch upon:

Does the discovery of permafrost make you rethink the pedo/geomorphology of the Pirin mountains?

Does it imply that we need to rethink the importance of permafrost formation factors?

Can permafrost be more common in Bulgaria than we previously thought? If that is the case, could it be that we are underestimating the permafrost distribution worldwide?

**Specific comments**

**Abstract**

You use the spelling "glaciarret". One of the other reviewers used the spelling "glaciaret". In Ganchev, (2011) I found it spelled as "glacieret". What is the correct spelling? Please determine which one is appropriate and use it throughout the manuscript.

**1. Introduction**

Line 36: "also" / in addition of what? Please remove "also".

Line 37: how is it visible? This is a rather vague sentence.

Line 51-53: references required

Line 54: "The ERT technique is one of the basic methods for permafrost evidence and studies". I don't know if basic is the best term here. A commonly used geophysical method maybe? Also, please add references.

Line 81: Electrical Resistivity Tomography (ERT) method

Line 81: is sensitive to the changes …

Line 85: significant how? Please be specific e.g. "increases by 'x' orders of magnitude", include a reference.

Line 86: Please delete this sentence. Either include this mention earlier in the paragraph or merge it with another sentence. On its own at the end of the paragraph doesn't fit well with how the text flows.

Line 93: Did you want to say: "even though"? I think what you say before "there are some limitations" is redundant.

Line 95: please rephrase as "the complicated logistics of transporting … and the aspects regarding a safe system of work"

Line 98-99: Not clear. Please rephrase.

Line 101: "To minimize the inaccuracies the relief can be estimated using unmanned aerial vehicle (UAV) for
example." Include a reference please.

Line 103: ERT has limitations when applied

Line 104: Use "firstly".

> Consequence of what?

> Ground not "rock"

Line 93-108: I can see that the point of this paragraph is to justify the complementary use of GPR, ERT and UAV measurements. However, you didn't say how exactly they benefit from each other. Please be a bit more specific. For example, ERT resulting subsurface images suffer inherently from non-uniqueness. GPR, as an independent method of subsurface investigation, can supply more accurate information about where the boundaries between layers are located. This in turn can be used to give some geological context to the ERT tomograms or even constrain the ERT inversion results.

Line 109: "a very short"

> What is a surface capture? Maybe I am not familiar with the term. Is it a series of photographs of the ground surface?

"An UAV"

Line 112-113: rephrase as "there is not much information about the ice thickness"

Please delete "and one point of our work was to solve this". You already said this above.

Line 115: Please delete what is in parenthesis.

Line 116: Move this sentence at the beginning of the paragraph where you state what you will be showing us in this manuscript, such as: "the aim was to investigate the thickness…the glacial bed). In addition, we investigated where the meltwater from Snezhnika disappears beneath the microglaciers' bed.

Line 117: Last sentence is redundant.

**2. Methods**

**2.1. Study site description**

Line 134: Rephrase "In the Golyam Kazan Cirque we find low coverage alpine and subalpine vegetation."

Line 135: "There is no evidence of surface water"

Line 136: This sentence is not well integrated in the text: "Pirin Mountain is on cross-roads of Mediterranean and Continental climate". As it stands, it sounds redundant.

Line 138: Is this correct: "measured until 2005"? 2005 was 17 years ago, I am sure a lot has change in the meantime. Are these records still representative?

**2.2. Ground-penetrating radar**

Table 1: GPR measurements settings

SURVEY |space| PARAMETERS

Line 165: What does this mean "without additional security? ". I think you mean that you didn't have the right gear, but you don't need to justify that. Just say: "highest elevation area accessible with the gear and tools available at the time."

Line 166: Change "the second reason" to " In addition, most of…"

**2.3. Electrical resistivity method**

Line 186: Do you mean "safe access to…"?

Line 186: why was pole-dipole and dipole-dipole impossible? You have to explain if you decide to mention it.

Line 191-194: This bit of text still doesn't sound right. I have given you a nice example in the previous round of revision, but you chose not to correct it.

> What is a real geoelectrical section? I think you mean the result you get post inversion, but it's the first time I come across this terminology. Please rephrase.

Line 198: rephrase to "harsh weather conditions did not allow us to work on the same date every year"

Line 201: rephrase to "This measurement window is relative, changing from one year to another, with 2017 and 2018 seeing the first snow fall in September, whereas in 2020, with a warm month of October, the first snow fall was recorded in []"

Figure 3. How did you obtain the outline of the glaciaret? Is that from UAV images/ photos? Please specify in text.

**3. Results and discussion**

This section as a whole does not have a lot of structure. I suggest the introduction of some subsections to aid the reader.

Line 210-211: I have no idea what this sentence means "and accordingly its lowest part consists of the" new" snow left from the last winter. This are profiles GPR(2018)-1, GPR(2018)-2 and partially GPR(2018)-3 (Fig. 4a,b and c)." Why is new in quotation marks? Do you mean those profiles are relevant because they are down the slope? Please rephrase and be clear.

Line 246: What depth of 11m? Is that what Gruenewald found? What aerial geophysical measurements? Is UAV photography a geophysical method? Please be clear. Rephrase.

Line 254: How is this sentence relevant here? - "Subsurface structure of microglacier bed was investigated using ERT and GPR measurements." Isn't this a generic sentence? How is it connected to the rest of the text?

Line 257: aren't the years implied by the survey names?

Fig 6 is barely discussed. Please expand the discussion around it or remove it.

Fig 7: You call those pseudosections in the text. Results obtained post inversions are not pseudosections. Pseudosections are a plot of apparent resistivity values, measurements obtained in the field.

An RMS error of 22.5 is quite high. Why is that? Don't you think this error might influence the validity of your conclusions regarding Fig 7c?

Line 259: It is definitely not just marble that can have resistivity values between 8000 and 40000 Ohm.m. Many mixtures of rock fragments, water and soil can have a resistivity response within the same resistivity range. Given the local lithology, I understand why you made such statement, but the phrase needs rephrasing. Presently, it sounds as if because it has that resistivity it must be marble, which is not true.

Line 284-286: Are you not just repeating what you said above?

Line 289: Figure numbler not included.

Line 300: Figure number not included.

Line 327-328: I don't understand this sentence. Please rephrase.

At the end of section 3 (and consequently at the end of the conclusion section) I would suggest including a short discussion about the wider implications of your study. Why is this work relevant/how is it useful for the study of permafrost/ geophysical methodology or geomorphology of Bulgarian mountains? How can future research build upon your results?

**4. Conclusions**

Line 332: rephrase "is a large-scale assessment of the ice-thickness"

Line 338: "by size"? do you mean surface area?

Line 342-343: "ERT measurements were repeated over three consecutive years, detecting the anomaly during every measurement campaign."

Line 344: rephrase "an answer to the question: where does the meltwater disappear?"

I would have my reservations about claiming you found the answer to that question. I feel that the evidence you present is not sufficient to claim that. You might however say that based on your observations you have formulated a new hypothesis.

**References**

Gachev, E. (2011). Inter-annual size variations of Snezhnika glacieret (the Pirin mountains, Bulgaria) in the last ten years. *Studia Geomorphologica Carpatho-Balcanica*, *45*(October), 7–19.

---

## Author Response (AR2)

Reply to Referee1, Mihai Cimpoiasu

This is my second review of this manuscript, now entitled: "Geophysical measurements of the Southernmost microglacier in Europe suggest permafrost occurrence in Bulgaria".

I acknowledge the clear improvements done to the article. I can safely say it is a valuable piece of science and it will be a good contribution to the literature. However, still needs a bit more work before it can be published. At the moment, I cannot recommend it for publication.

First of all, the language issue still persists especially in the paragraphs where new text was added. There are still quite a lot of mistakes and several sentences difficult to understand. I have tried to help the authors with some suggestions as much as I could, but the whole text needs to be thoroughly examined and its language more rigorously improved.

**Thank you for the effort to correct the language. The final correction of the manuscript is made by an English teacher. We hope the text is better readable now.**

There is a serious lack of detail about how ERT measurements were collected. You don't even mention the device used.

**The device used for ERT measurements is ABEM SAS 1000. It is mentioned in the text on line 173.**

How many measurements per profile?

**24 electrodes allow 121 measurements per profile. We added this information to the text.**

*… The multi-electrode resistivity technique consists in using a multi-core cable with 24 conductors, as electrodes plugged into the ground at a fixed spacing. The number of 24 electrodes allowed 121 measurements per profile…*

Did you acquire any reciprocal data?

**Measurements were conducted in forward and reciprocal directions and the final values are the average of the resistivity at each point, measured in both directions. This is the usual way of collecting data from ERT in Bulgaria and we have not mentioned it in the text.**

Any data filtering applied pre-inversion?

**Yes, the filtering was performed before the inversion. A small paragraph is added to the text describing the workflow.**

*Obtained ERT data was processed with RES2DINV (Loke 2001, 2010). Data preprocessing include extermination of bad data points and applying of vertical/horizontal filter weight. An option for effect reduction of side blocks and obtaining smooth anomalies is also used. First a trial inversion is made and after it a RMS (Root Mean Square) error statistics is performed. Then the data points with extreme values are removed and the inversion is made again.*

Any error models applied to the data?

**Error model was made and the data was corrected based on it.**

Any temperature corrections?

**We didn't apply temperature corrections because the anomalies caused by the ice lenses and ice rich permafrost have a big difference in the resistivity values (more than 60 000 $\Omega m$) than the accommodating medium. According to the Hauck (2001), there is an exponential change in the resistivity with the decrease of the temperature below 0.**

*Hauck (2001) Geophysical methods for detecting permafrost in high mountains, Versuchsanstalt fur Wasserbau Hydrologie und Glaziologie der Eidgenossischen Technischen Hochschule Zurich, 2001(p.73-76)*

It looks like you used Res2DInv as an inversion software, but you are not referencing it. Why is that?

**Thank you for this remark, we added the name of the software and the reference in the text.**

Finally, it would be useful to discuss the wider implications of your work. Why are your results relevant to the wider scientific community? Examples of a few discussion topics you might want to touch upon:

Does the discovery of permafrost make you rethink the pedo/geomorphology of the Pirin mountains?

Does it imply that we need to rethink the importance of permafrost formation factors?

Can permafrost be more common in Bulgaria than we previously thought? If that is the case, could it be that we are underestimating the permafrost distribution worldwide?

**Thank you for this suggestion. The questions you ask are very interesting. We have discussed shortly the presence of permafrost in Pirin Mountain but pedogenesis is a topic, which is a bit far from ours. The following paragraphs are added at the end of Sections 3 and 4:**

*Bulgaria and the Balkan Peninsula are situated in lower latitudes where no continuous permafrost exists. Only isolated patches of permafrost are present in high mountains (Brown et al.(2001)). However, its distribution is much less investigated (Oliva et al.(2018)). Permafrost probably exists above ∼ 2350 m on Rila Mountains (Oliva et al.(2018)) and above ∼ 2400 m in the Pirin Mountains. In the Golyam Kazan cirque is can be sporadic and presented in the area south of the glacieret, where snow patches are observed in some summers. To prove this more studies including geophysical measurements are needed. The area was described above as well shaded by the Dzhamdzhiev ridge. Most of the places in the high mountains of Bulgaria, where probably permafrost is present, are steep and this makes it difficult for geophysical measurements to be carried out. Nevertheless, more investigation into permafrost distribution and its monitoring in time should be made. It is important because changing climatic conditions can lead to the disappearing of isolated permafrost patches especially in areas with relatively warm climate like the Balkan Peninsula. Disappearing of permafrost patches can cause rapid degradation processes and rock avalanches. In Bulgaria there are no many infrastructures in the high mountains that can be damaged, unlike in the Alps or other inhabited mountains. However, many hike trails are crossing slopes that can be unstable and dangerous when permafrost melts.*

*The knowledge of permafrost on the Balkan Peninsula and particularly in Bulgaria is very general and studies of permafrost, including geophysical measurements, are still rare. Even though the presented study is focused on a small area of Pirin mountain (Bulgaria), it gives important information on the presence, extent, and the state of a permafrost area, surviving due to local conditions in the warming climate.*

Specific comments

Abstract

You use the spelling "glaciarret". One of the other reviewers used the spelling "glaciaret". In Ganchev, (2011) I found it spelled as "glacieret". What is the correct spelling? Please determine which one is appropriate and use it throughout the manuscript.

**Thank you for the comment, we unified the spelling as "glacieret according to Ganchev, (2011).**

1. Introduction Line 36: "also" / in addition of what? Please remove "also".

**Done**

Line 37: how is it visible? This is a rather vague sentence.

**We decided to remove the sentence.**

Line 51-53: references required

**References are added.**

Line 54: "The ERT technique is one of the basic methods for permafrost evidence and studies". I don't know if basic is the best term here. A commonly used geophysical method maybe? Also, please add references.

**The proposed change is made and a reference is added.**

Line 81: Electrical Resistivity Tomography (ERT) method

**Done**

Line 81: is sensitive to the changes ...

**Done**

Line 85: significant how? Please be specific e.g. "increases by 'x' orders of magnitude", include a reference.

**The change is exponential. It is added to the text together with a reference.**

Line 86: Please delete this sentence. Either include this mention earlier in the paragraph or merge it with another sentence. On its own at the end of the paragraph doesn't fit well with how the text flows.

**The sentence is merged with the first one from the paragraph. Thank you for the suggestion.**

Line 93: Did you want to say: "even though"? I think what you say before "there are some limitations"is redundant.

**The sentence is changed as suggested.**

Line 95: please rephrase as "the complicated logistics of transporting ... and the aspects regarding a safe system of work"

**Done**

Line 98-99: Not clear. Please rephrase.

**Done**

Line 101: "To minimize the inaccuracies the relief can be estimated using unmanned aerial vehicle (UAV) for example." Include a reference please.

**References are added.**

Line 103: ERT has limitations when applied

**Done**

Line 104: Use "firstly". Consequence of what? Ground not "rock"

**The sentence is corrected as suggested.**

Line 93-108: I can see that the point of this paragraph is to justify the complementary use of GPR, ERT and UAV measurements. However, you didn't say how exactly they benefit from each other. Please be a bit more specific. For example, ERT resulting subsurface images suffer inherently from non-uniqueness. GPR, as an independent method of subsurface investigation, can supply more accurate information about where the boundaries between layers are located. This in turn can be used to give some geological context to the ERT tomograms or even constrain the ERT inversion results.

**To add such a paragraph, describing limitations of the methods, was suggested by the Referee2. We have used the text you propose to extend the paragraph a little bit.**

Line 109: "a very short" What is a surface capture? Maybe I am not familiar with the term. Is it a series of photographs of the ground surface? "An UAV"

**Yes, we mean under "surface capture" a series of photographs, made by the drone. They are aligned and processed after that to produce a model of the surface.**

Line 112-113: rephrase as "there is not much information about the ice thickness" Please delete "and one point of our work was to solve this". You already said this above.

**Done**

Line 115: Please delete what is in parenthesis.

**Done**

Line 116: Move this sentence at the beginning of the paragraph where you state what you will be showing us in this manuscript, such as: "the aim was to investigate the thickness...the glacial bed). In addition, we investigated where the meltwater from Snezhnika disappears beneath the microglaciers' bed.

**The suggested change is made.**

Line 117: Last sentence is redundant.

**The sentence is removed.**

2. Methods

2.1. Study site description

Line 134: Rephrase "In the Golyam Kazan Cirque we find low coverage alpine and subalpine vegetation."

**Done**

Line 135: "There is no evidence of surface water"

**Done**

Line 136: This sentence is not well integrated in the text: "Pirin Mountain is on cross-roads of Mediterranean and Continental climate". As it stands, it sounds redundant.

**We changed the order of the sentences in the paragraph. Now it starts with the climate, then vegetation, and at the end is the sentence about the surface water.**

Line 138: Is this correct: "measured until 2005"? 2005 was 17 years ago, I am sure a lot has change in the meantime. Are these records still representative?

**Yes, you are right. The records were made 17 years ago, but are the most recent official information we have found. According to our observations during winter hikes in Pirin (not related to the research), the snow at the end of winter is 2-3m. It can be easily appreciated, since the snow completely covers the dwarf mountain pines (Pinus mugo), which in Rila and Pirin Mountains reaches about 2 m.**

2.2. Ground-penetrating radar

Table 1: GPR measurements settings
SURVEY |space| PARAMETERS

**Done**

Line 165: What does this mean "without additional security? ". I think you mean that you didn't have the right gear, but you don't need to justify that. Just say: "highest elevation area accessible with the gear and tools available at the time."

**Thank you for the suggestion, we changed the text.**

Line 166: Change "the second reason" to " In addition, most of..."

**Done**

2.3. Electrical resistivity method

Line 186: Do you mean "safe access to..."?

**Yes, "safe" is the right word. It is corrected now.**

Line 186: why was pole-dipole and dipole-dipole impossible? You have to explain if you decide to mention it.

**We decided to remove this part of the sentence. Pole-dipole and dipole-dipole measuring schemes were not possible to be used due to the rough terrain.**

Line 191-194: This bit of text still doesn't sound right. I have given you a nice example in the previous round of revision, but you chose not to correct it. What is a real geoelectrical section? I think you mean the result you get post inversion, but it's the first time I come across this terminology. Please rephrase.

**Thank you for the suggestion. We corrected the text according to your first suggestion.**

Line 198: rephrase to "harsh weather conditions did not allow us to work on the same date every year"

**Thank you for the suggestion, we have accepted it and changed the text accordingly.**

Line 201: rephrase to "This measurement window is relative, changing from one year to another, with 2017 and 2018 seeing the first snow fall in September, whereas in 2020, with a warm month of October, the first snow fall was recorded in []"

**Thank you for the suggestion, the sentence is rephrased.**

Figure 3. How did you obtain the outline of the glaciaret? Is that from UAV images/ photos? Please specify in text.

**The outlines of the microglacier were recorded with Garmin and for 2018 it was also improved with the UAV image. The information is added in the main text of the manuscript. Thank you for this remark.**

3. Results and discussion

This section as a whole does not have a lot of structure. I suggest the introduction of some subsections to aid the reader.

**There were subsections in the first version but we decided to remove them. Now we uncommented the rows with subsections and changed the names. Thank you for this suggestion.**

Line 210-211: I have no idea what this sentence means "and accordingly its lowest part consists of the" new" snow left from the last winter. This are profiles GPR(2018)-1, GPR(2018)-2 and partially GPR(2018)-3 (Fig. 4a,b and c)." Why is new in quotation marks? Do you mean those profiles are relevant because they are down the slope? Please rephrase and be clear.

**Thank you for the comment, we changed these sentences to be more clear.**

*...In 2018 the size of microglacier was bigger than in 2017 (Fig.2a and Fig.3) and accordingly its lowest part consists only of the snow left from the last winter. This can be observed on profiles GPR(2018)-1, GPR(2018)-2 and partially GPR(2018)-3 (Fig.4a,b and c) situated in the lowest part of the microglacier. Within the first layer, there are some less differentiated discontinuities...*

Line 246: What depth of 11m? Is that what Gruenewald found? What aerial geophysical measurements? Is UAV photography a geophysical method? Please be clear. Rephrase.

**Thank you for the comment, the text in this paragraph is changed to be more clear.**

Line 254: How is this sentence relevant here? - "Subsurface structure of microglacier bed was investigated using ERT and GPR measurements." Isn't this a generic sentence? How is it connected to the rest of the text?

**The sentence is merged with the next one.**

Line 257: aren't the years implied by the survey names?

**We removed the years in the name of the profile only here due to the repetition.**

Fig 6 is barely discussed. Please expand the discussion around it or remove it.

**You are right, after the last editing we have removed some text connected with this figure. Now we decided to remove also the figure.**

Fig 7: You call those pseudosections in the text. Results obtained post inversions are not pseudosections. Pseudosections are a plot of apparent resistivity values, measurements obtained in the field. An RMS error of 22.5 is quite high. Why is that? Don't you think this error might influence the validity of your conclusions regarding Fig 7c?

**Thank you for this remark. The wrong plot/not the last one was merged in this figure. We check the figures before submission but the difference is very small and we missed it. Now the correct one is merged.**

Line 259: It is definitely not just marble that can have resistivity values between 8000 and 40000 Ohm.m. Many mixtures of rock fragments, water and soil can have a resistivity response within the same resistivity range. Given the local lithology, I understand why you made such statement, but the phrase needs rephrasing. Presently, it sounds as if because it has that resistivity it must be marble, which is not true.

**The correction is made.**

Line 284-286: Are you not just repeating what you said above?

**The text is reordered and the repetition is corrected.**

Line 289: Figure numbler not included.

**Figure number is included.**

Line 300: Figure number not included.

**Figure number is included.**

Line 327-328: I don't understand this sentence. Please rephrase.

**Done**

At the end of section 3 (and consequently at the end of the conclusion section) I would suggest including a short discussion about the wider implications of your study. Why is this work relevant/how is it useful for the study of permafrost/ geophysical methodology or geomorphology of Bulgarian mountains? How can future research build upon your results?

**Two short paragraphs are added as suggested.**

4. Conclusions Line 332: rephrase "is a large-scale assessment of the ice-thickness"

**Done**

Line 338: "by size"? do you mean surface area?

**Yes, we mean "be the surface area". The correction is made.**

Line 342-343: "ERT measurements were repeated over three consecutive years, detecting the anomaly during every measurement campaign."

**Done**

Line 344: rephrase "an answer to the question: where does the meltwater disappear?" I would have my reservations about claiming you found the answer to that question. I feel that the evidence you present is not sufficient to claim that. You might however say that based on your observations you have formulated a new hypothesis.

**The correction is made**

References
Gachev, E. (2011). Inter-annual size variations of Snezhnika glacieret (the Pirin mountains, Bulgaria) in the last ten years. Studia Geomorphologica Carpatho-Balcanica, 45(October), 7–19

Reply to Anonymous Referee2

I have only several specific comments listed below: - Title: I suggest keeping 'Pirin Mountains' in the title and placing 'Bulgaria' in the bracket.

**The title is changed according to suggestions**

- You should decide if your paper deals with perennial snow patches or glaciers. Despite, you having changed the title there are still many phrases in which you are presenting the importance of snow patches etc. In addition, you make confusion when considering snow patches as remnants of glaciers.

**The term "perennial snow patches" has been removed in most of the places yet. But we decided to keep it in some sentences in the Introduction.**

- Replace 'glaciarret' with 'glacieret' (see Gachev's papers).

**Done**

- Language should be improved!

**The final correction is made by an English teacher. We hope the text is better readable now.**

Line 14: delete the definition of 'perennial snow patches' because in the title (and the entire manuscript) you present a glacieret/microglacier.

**Done**

Line 23: replace 'since the Holocene' with 'since the end of the Pleistocene'

**Replaced**

Line 25: again perennial snow patch? You should decide if you discuss about snow patches or glaciers!

**We have removed "perennial snow patches" in most places but some we decided to keep.**

Line 27: when had Vihren glacier that size? LGM or LIA?

**Thank you for the comment. The given size is of the microglacier between 1994 and now. The sentence is rewritten to be correct.**

Lines 37-45: I think this part should be improved. 'In my opinion, you should refer here to the relation between permafrost and glaciers.

**Thank you for the comment. A few sentences about the permafrost-glacier interactions are added to the text.**

*Glaciers and permafrost are well studied separately but the interaction between them is topic of less publications (Harris et al.(2005))......Permafrost can be absent in areas where glaciers are present but the ice rich permafrost and ground ice can be formed in front of and beneath the glaciers (Harris et al.(2005))....*

Lines 63-64 and elsewhere: when citing more papers try to list them ordered by the year.

**Thank you for the comment, the references are ordered yet.**

Lines 64-65: what do you mean? Please rephrase! In the case of seismic, it is not true!

**"Geophysical methods" is replaced with "GPR and ERT". Thank you for this remark.**

Line 84: 'The perennial snow patches in high mountains are similar to mountain glaciers (as remnants of them)'. Snow is snow and ice is ice (glacier), don't mix it! You make confusion here. Please eliminate this phrase! Snow patches are not remnants of glaciers! Snow patches are the accumulation of snow. Remnants of glaciers are glacierets, ice patches, ice masses etc

**We have removed the suggested part of the sentence.**